# Understanding Matrix Function Normalizations in Covariance Pooling through the Lens of Riemannian Geometry

**Ziheng Chen**[1]*, **Yue Song**[1]*, **Xiao-Jun Wu**[2], **Gaowen Liu**[3] **& Nicu Sebe**[1]
[1] University of Trento, [2] Jiangnan University, [3] Cisco Systems
`ziheng_ch@163.com, yue.song@unitn.it`

## Abstract

Global Covariance Pooling (GCP) has been demonstrated to improve the performance of Deep Neural Networks (DNNs) by exploiting second-order statistics of high-level representations. GCP typically performs classification of the covariance matrices by applying matrix function normalization, such as matrix logarithm or power, followed by a Euclidean classifier. However, covariance matrices inherently lie in a Riemannian manifold, known as the Symmetric Positive Definite (SPD) manifold. The current literature does not provide a satisfactory explanation of why Euclidean classifiers can be applied directly to Riemannian features after the normalization of the matrix power. To mitigate this gap, this paper provides a comprehensive and unified understanding of the matrix logarithm and power from a Riemannian geometry perspective. The underlying mechanism of matrix functions in GCP is interpreted from two perspectives: one based on tangent classifiers (Euclidean classifiers on the tangent space) and the other based on Riemannian classifiers. Via theoretical analysis and empirical validation through extensive experiments on fine-grained and large-scale visual classification datasets, we conclude that the working mechanism of the matrix functions should be attributed to the Riemannian classifiers they implicitly respect. The code is available at https://github.com/GitZH-Chen/RiemGCP.git.

## 1 Introduction

Global Covariance Pooling (GCP), a method used as a replacement for Global Average Pooling (GAP) in aggregating the final activations of Deep Neural Networks (DNNs), has demonstrated exceptional performance improvements across various applications (Lin et al., 2015; Ionescu et al., 2015; Li et al., 2017; Wang et al., 2017; Koniusz et al., 2017; Li et al., 2018; Wang et al., 2020a; Rahman et al., 2020; Zhu et al., 2024). The research line of existing GCP methods mainly focuses on improving performance by adopting different normalization methods (Ionescu et al., 2015; Li et al., 2017; Wang et al., 2020a), exploiting richer statistics (Cui et al., 2017; Wang et al., 2017; Koniusz et al., 2021; Rahman et al., 2023; Chen et al., 2023a), improving covariance conditioning (Song et al., 2022d;a), and obtaining compact representations (Gao et al., 2016; Yu & Salzmann, 2018; Lin et al., 2018; Wang et al., 2022a). Generally, a GCP meta-layer computes the covariance matrix of the activations as the global representation, and then performs normalization either by *matrix logarithm* (Ionescu et al., 2015) or *matrix power* (Li et al., 2017; 2018; Wang et al., 2020a). Finally, the normalized matrices are fed into a Euclidean classifier. The square root has emerged as the most effective normalization scheme, outperforming the logarithm counterpart by a large margin (Li et al., 2017; Wang et al., 2020a; Song et al., 2021). Although the research community has provided some theoretical support for the matrix logarithm, there are no intrinsic explanations for the matrix power.

The covariance matrices naturally lie in a Riemannian manifold, known as Symmetric Positive Definite (SPD) manifolds (Pennec et al., 2006). For matrix logarithm, it maps SPD matrices into the Euclidean space of the tangent space at the identity matrix. Euclidean classifiers can, therefore, be applied after the matrix logarithm. However, the co-domain of the matrix power is still the SPD

---
*Corresponding author

manifold, rendering the application of Euclidean classifiers following matrix power less mathematically supported. Several works have attempted to explain the matrix power. The initial motivation of matrix power in GCP (Li et al., 2017) is that the distance induced by the matrix square root approximates the geodesic distance under Log-Euclidean Metric (LEM) (Dryden et al., 2010). Nevertheless, Fig. 1 shows that the gap between these two distances is still noticeable. Furthermore, Song et al. (2021) empirically explored the benefits of approximate matrix square root over its accurate counterpart, while Wang et al. (2020b) studied the merits of GCP from an optimization perspective. *However, none of them touch upon the fundamental reason why Euclidean classifiers can be directly employed in the non-Euclidean co-domain of the matrix power.* There appears to be a discrepancy between theoretical principles and practical applications of matrix power and logarithm.

This study aims to offer a comprehensive theoretical understanding of the matrix logarithm/power in GCP and reconcile the discrepancy between theory and practice. Without loss of generality, we refer to matrix logarithm and power collectively as matrix functions. Given that the matrix logarithm is a Riemannian logarithmic map, mapping SPD data into the tangent space, we first systematically study Riemannian logarithmic maps on SPD manifolds under seven families of metrics, resulting in three types of Riemannian logarithmic maps, the ones based on the matrix logarithm, matrix power, and Log-Cholesky Metric (LCM), respectively. Consequently, the matrix logarithm in GCP establishes a tangent classifier (Euclidean classifiers on the tangent space) for covariance classification. Also, by applying a simple affine transformation, the matrix power in GCP constructs a tangent classifier. This indicates that we might unify both matrix logarithm and power as tangent classifiers. However, our experiments suggest that this tangent classifier explanation fails to account for the efficacy of matrix power. As the tangent space distorts the intrinsic geometry of manifolds, we conjecture that tangent classifiers might not be the underlying mechanisms.

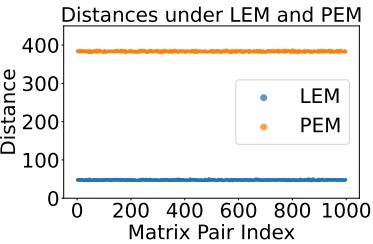

Figure 1: The metric induced by the matrix power is the Power Euclidean Metric (PEM). Although PEM approaches LEM as the power approaches 0, the distances under **PEM** ($\theta = 0.5$) and **LEM** still differ largely. We visualize these two distances for 1000 random pairs of $256 \times 256$ SPD matrices. The average difference is $335.84 \pm 1.61$. This indicates that PEM is not proximate to LEM under the widely used $\theta = 0.5$.

To delve further, we move on to a more intrinsic explanation based on the recently developed SPD Multinomial Logistics Regression (MLR) (Nguyen & Yang, 2023; Chen et al., 2024a;d), which extends the Euclidean MLR (FC + softmax) into manifolds. Based on previous work (Chen et al., 2024a), we find that matrix logarithm in GCP implicitly constructs the SPD MLR under LEM. Furthermore, we theoretically demonstrate that the matrix power in GCP implicitly respects the SPD MLR under PEM. These findings suggest that matrix functions in GCP can be uniformly interpreted as Riemann classifiers. Therefore, the observed performance gap between the matrix power and logarithm can be attributed to the characteristics of the underlying Riemannian metrics. To validate this postulation, we conduct experiments on the ImageNet-1k (Deng et al., 2009) and three Fine-Grained Visual Categorization (FGVC) datasets, namely Caltech University Birds (Birds) (Welinder et al., 2010), Stanford Cars (Cars) (Krause et al., 2013), and FGVC Aircrafts (Aircrafts) (Maji et al., 2013). *The results confirm that the Riemannian classifier rather than the tangent classifier contributes to the efficacy of matrix functions in GCP.* We expect our work to pave the way for a deeper theoretical understanding of GCP from a Riemannian perspective and inspire more research to explore the rich SPD geometries for more effective GCP applications. We present a teaser table in Tab. 1. Due to page limits, we put the related work in App. B and all the proofs in the appendix. Besides, tables of notations and abbreviations are presented in App. C for better readability.

In summary, our main **contributions** are two-fold. **(a). First intrinsic explanation for matrix normalization.** We explain the working mechanism of matrix functions in GCP from the perspectives of tangent and Riemannian classifiers, and finally claim that the rationality of matrix functions should be attributed to the Riemannian classifiers they implicitly respect. To the best of our knowledge, this is the first Riemannian interpretation of the matrix functions in GCP. **(b). Empirical validation by extensive experiments.** We validate our theoretical argument on large-scale and FGVC datasets.

Table 1: **Main results:** The working mechanisms of matrix functions in GCP are attributed to Riemannian classifiers they implicitly respect.

| Matrix function | Intrinsic explanation | Used in GCP | Reference |
|---|---|---|---|
| Logarithm | LEM-induced Riemannian Classifier | Log-EMLR (Eq. (4)) | (Chen et al., 2024a, Prop. 5.1) |
| Power | PEM-induced Riemannian Classifier | Pow-EMLR (Eq. (5)) | Thm. 2 |

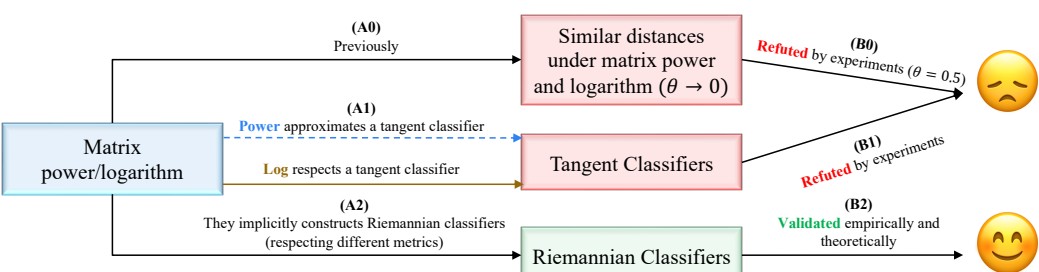

Figure 2: Illustration on the main postulations (A0 to A2) and empirical validations (B0 to B2) of our investigation, where $\theta$ is the power in the matrix power. Postulation **A0** is adopted by Li et al. (2017) and is refuted by our experiments in Fig. 1 for the specific $\theta = 0.5$. Postulation **A1** is indicated by Tab. 2 and is refuted by our experiments in Sec. 6. Postulation **A2** is supported by Thm. 2 and is validated by our experiments in Sec. 6.

**Main theoretical results:** Tab. 2 presents a complete list of Riemannian logarithmic maps on SPD manifolds under different metrics. It indicates that the matrix power, with a simple affine transformation, can serve as a Riemannian logarithmic map. Since the matrix logarithm has been widely recognized as a building component of tangent classifiers, we also expect that the matrix power function can be explained by tangent classifiers. However, the preliminary experiments presented in Tab. 3 refute this conjecture, suggesting the existence of more fundamental mechanisms. Therefore, we delve into this mystery in Sec. 5 by leveraging the recently developed Riemannian classifiers. *Thm. 2 indicates that matrix power in GCP implicitly establishes a Riemannian classifier for covariance matrix classification.* Similar results also hold for the matrix logarithm (Chen et al., 2024a). This implies that the matrix logarithm and power can be unifiedly interpreted as essential components of Riemannian classifiers. Tab. 4 summarizes all our theoretical findings. Sec. 6 further validate our theoretical explanations by extensive experiments. The reasoning behind our analysis is illustrated in Fig. 2.

## 2 THE GEOMETRY OF SPD MANIFOLDS

Let $\mathcal{S}_{++}^n$ be the set of $n \times n$ SPD matrices. As shown by Arsigny et al. (2005), $\mathcal{S}_{++}^n$ is an open submanifold of the Euclidean space $\mathcal{S}^n$ of symmetric matrices. There are five popular Riemannian metrics on SPD manifolds: Affine-Invariant Metric (AIM) (Pennec et al., 2006), Log-Euclidean Metric (LEM) (Arsigny et al., 2005), Power-Euclidean Metric (PEM) (Dryden et al., 2010), Log-Cholesky Metric (LCM) (Lin, 2019), and Bures-Wasserstein Metric (BWM) (Bhatia et al., 2019). Note that when power equals 1, PEM reduces to the Euclidean Metric (EM). All of the above five standard metrics have been generalized into parameterized families of metrics.

Thanwerdas & Pennec (2023) generalized AIM, LEM, and EM into two-parameter metrics by the $O(n)$-invariant Euclidean inner product on $\mathcal{S}^n$:

$$\langle V, W \rangle^{(\alpha, \beta)} = \alpha \langle V, W \rangle + \beta \operatorname{tr}(V) \operatorname{tr}(W), \tag{1}$$

where $\alpha > 0$ and $\beta > -\alpha/n$, $V, W \in \mathcal{S}^n$, and $\langle \cdot, \cdot \rangle$ is the standard matrix inner product. These generalized metrics are denoted as $(\alpha, \beta)$-AIM, $(\alpha, \beta)$-LEM, and $(\alpha, \beta)$-EM, respectively. Besides, Thanwerdas & Pennec (2022) defined the power-deformed metric $\tilde{g}$ of a metric $g$ on $\mathcal{S}_{++}^n$ as

$$\tilde{g}_P (V, W) = \frac{1}{\theta^2} g_{P^\theta} \left( (\operatorname{Pow}_\theta)_{*,P}(V), (\operatorname{Pow}_\theta)_{*,P}(W) \right), \forall P \in \mathcal{S}_{++}^n, V, W \in T_P \mathcal{S}_{++}^n, \tag{2}$$

where $\operatorname{Pow}_\theta(P) = P^\theta$ denotes matrix power, $(\operatorname{Pow}_\theta)_{*,P}$ is the differential map of $\operatorname{Pow}_\theta$ at $P$, and $T_P \mathcal{S}_{++}^n$ is the tangent space at $P$. Following Eq. (2), $(\alpha, \beta)$-AIM, $(\alpha, \beta)$-LEM, $(\alpha, \beta)$-EM, LCM,

and BWM are generalized into power-deformed families of metrics, denoted as $(\theta, \alpha, \beta)$-AIM, $(\theta, \alpha, \beta)$-LEM, $(\theta, \alpha, \beta)$-EM, $\theta$-LCM, and $2\theta$-BWM, respectively (Thanwerdas & Pennec, 2022; Chen et al., 2024d). Chen et al. (2024d) further shows that $(\theta, \alpha, \beta)$-LEM equals $(\alpha, \beta)$-LEM. Besides, as shown by Thanwerdas & Pennec (2022); Chen et al. (2024d), $\theta$ serves as a deformation from a LEM-like metric. For instance, $(\theta, \alpha, \beta)$-AIM becomes $(\alpha, \beta)$-AIM when $\theta = 1$ and approaching $(\alpha, \beta)$-LEM with $\theta \to 0$.

On the other hand, PEM was generalized into Mixed Power Euclidean Metric (MPEM) (Thanwerdas & Pennec, 2019) by two power factors $\theta_1$ and $\theta_2$, denoted as $(\theta_1, \theta_2)$-EM. When $\theta_1 = \theta_2$, MPEM is reduced to PEM. Han et al. (2023) extended BWM into Generalized Bures-Wasserstein Metric (GBWM) by an SPD parameter $M$, denoted as $M$-BWM. When $M = I$, GBWM is reduced to BWM. We further generalize GBWM into $(2\theta, M)$-BWM by power deformation Eq. (2).

In total, three parameters $(\theta, \alpha, \beta)$ are involved in the metrics on the SPD manifold. The power deformation $\theta$ characterizes deformation (Chen et al., 2024d; Thanwerdas & Pennec, 2022), while $(\alpha, \beta)$ characterizes the $O(n)$-invariance, a powerful property in modeling covariance (Thanwerdas & Pennec, 2023). The above metrics have shown success in various applications, due to their closed-form expressions for Riemannian operators, such as the Riemannian logarithmic and exponential maps. We summarize the involved Riemannian operators in App. D.2.

## 3 GLOBAL COVARIANCE POOLING REVISITED

GCP captures the second-order statistics of the features in the last layer of the deep network. The standard GCP procedure comprises calculating the covariance matrix, normalization with a matrix function, vectorization, dimensionality reduction by an FC layer, and ultimately applying a Euclidean classifier. The sequence of these operations can be represented as follows:

$$X \xrightarrow{\text{Cov}} \Sigma \xrightarrow{f_{\text{M}}} \tilde{\Sigma} \xrightarrow{f_{\text{vec}}} x \xrightarrow{f_{\text{FC}}} \tilde{x} \xrightarrow{f_{\text{EC}}} \hat{y}, \tag{3}$$

where $f_{\text{M}}$, $f_{\text{vec}}$, $f_{\text{FC}}$ and $f_{\text{EC}}$ denote the matrix function, vectorization, FC layer, and Euclidean classifier, respectively. Typical candidates of matrix functions are matrix power and logarithm. As softmax is the most widely used classifier, $f_{\text{EC}}$ always denotes the softmax in this paper. However, our discussions can also apply to other classifiers used in GCP, such as SVM (Li et al., 2017; Wang et al., 2020a), as other classifiers receive the FC features as their inputs.

FC + softmax is known as Euclidean Multinomial Logistics Regression (EMLR). When the matrix function is the matrix power, we call the process $f_{\text{EC}} \circ f_{\text{FC}} \circ f_{\text{vec}} \circ f_{\text{M}}$ as the Pow-EMLR, while the counterpart of matrix logarithm is referred to as Log-EMLR. Especially, setting power as $1/2$ in GCP normally reaches the optimal performance (Li et al., 2017, Fig. 3). The Pow-EMLR and Log-EMLR can be formally expressed as

$$\text{Log-EMLR: softmax}\left(\mathcal{F}\left(f_{\text{vec}}\left(\text{mlog}(S)\right); A, b\right)\right), \tag{4}$$

$$\text{Pow-EMLR: softmax}\left(\mathcal{F}\left(f_{\text{vec}}\left(S^{\theta}\right); A, b\right)\right), \tag{5}$$

where $\mathcal{F}(\cdot; A, b)$ denotes the FC layer with the transformation matrix $A$ and biasing vector $b$.

## 4 MATRIX FUNCTIONS AND TANGENT CLASSIFIERS

The matrix logarithm is the Riemannian logarithmic map at the identity matrix $I$, mapping SPD matrices into the tangent space $T_I \mathcal{S}_{++}^n \cong \mathcal{S}^n$. As tangent spaces are Euclidean spaces, it is natural to exploit FC and Euclidean classifiers on $T_I \mathcal{S}_{++}^n$ directly. We refer to the Euclidean classifiers over the tangent space at the identity matrix, $T_I \mathcal{S}_{++}^n$, as tangent classifiers. This section systematically studies all Riemannian logarithmic maps between $\mathcal{S}_{++}^n$ and $T_I \mathcal{S}_{++}^n$, under seven families of metrics.

### 4.1 RIEMANNIAN LOGARITHMS UNDER SEVEN DEFORMED METRICS

The matrix logarithm is generally characterized as the Riemannian logarithm $\text{Log}_I$ at $I$ under the standard LEM and AIM. Inspired by this, we systematically investigate Riemannian logarithms on SPD manifolds. Let $P$ denote an SPD matrix and $\tilde{L}$ represent the Cholesky factor of $P^{\theta}$. Tab. 2

Table 2: $\text{Log}_I$ under seven families of metrics. $\theta_0 = \frac{\theta_1+\theta_2}{2}$ for $(\theta_1, \theta_2)$-EM, $\theta_0 = \theta$ for $(\theta, \alpha, \beta)$-EM and $2\theta$-BWM, and $(2\theta, \phi_{2\theta}(P))$-BWM.

| Metric | $\text{Log}_I P$ | Metric | $\text{Log}_I P$ |
|---|---|---|---|
| $(\alpha, \beta)$-LEM $(\theta, \alpha, \beta)$-AIM | $\text{mlog}(P)$ | $(\theta, \alpha, \beta)$-EM $(\theta_1, \theta_2)$-EM $2\theta$-BWM $(2\theta, P^{2\theta})$-BWM | $\frac{1}{\theta_0}(P^{\theta_0} - I)$ |
| $\theta$-LCM | $\frac{1}{\theta}\left[\lfloor\tilde{L}\rfloor + \lfloor\tilde{L}\rfloor^\top + 2\,\text{Dlog}(\mathbb{D}(\tilde{L}))\right]$ | | |

presents the Riemannian logarithms at $I$ under all seven metrics, where $\lfloor\cdot\rfloor$ is the strictly lower triangular part of a square matrix, $\mathbb{D}(\cdot)$ is a diagonal matrix, and $\text{Dlog}(\cdot)$ is the diagonal logarithm. We leave technical details in App. E.

*Remark* 1. Let us elaborate further on the parameter of GBWM in Tab. 2. Given an SPD point $P \in \mathcal{S}^n_{++}$, $P$-BWM coincides with the standard AIM in the neighborhood of $P$ (Han et al., 2021). This local property could be beneficial (Han et al., 2023). Similarly, $(2\theta, P^{2\theta})$-BWM is locally $(2\theta, 1, 0)$-AIM, the deformed metric of the standard AIM. Please refer to App. F for technical details.

Tab. 2 implies that there are three types of $\text{Log}_I$:

$$\text{Matrix-logarithm-based: } \text{mlog}(P), \tag{6}$$

$$\text{Matrix-power-based: } \frac{1}{\theta}(P^\theta - I), \tag{7}$$

$$\text{LCM-based: } \frac{1}{\theta}\left[\lfloor\tilde{L}\rfloor + \lfloor\tilde{L}\rfloor^\top + 2\,\text{Dlog}(\mathbb{D}(\tilde{L}))\right], \tag{8}$$

We denote the tangent MLR induced by Eq. (6), *i.e.*, Eq. (6) + vectorization + FC + softmax, as Log-TMLR, while the counterparts of Eq. (7) and Eq. (8) is referred to as Pow-TMLR and Cho-TMLR, respectively. Obviously, Log-TMLR is the exact Log-EMLR (Eq. (4)) used in GCP.

Table 3: Results of GCP on the ImageNet-1k and Cars datasets with Pow-TMLR or Pow-EMLR under the architecture of ResNet-18.

| Method | ImageNet-1k | | Cars | |
|---|---|---|---|---|
| | Top-1 Acc (%) | Top-5 Acc (%) | Top-1 Acc (%) | Top-5 Acc (%) |
| Pow-TMLR | 71.62 | 89.73 | 51.14 | 74.29 |
| Pow-EMLR | **73** | **90.91** | **80.43** | **94.15** |

## 4.2 POW-TMLR VERSUS POW-EMLR

Pow-EMLR applies Euclidean MLR directly on the non-Euclidean SPD manifold, while Pow-TMLR applies Euclidean MLR on the Euclidean space of $T_I\mathcal{S}^n_{++}$. *In this sense, Pow-TMLR should be more theoretically advantageous than Pow-EMLR.* Moreover, the difference between Pow-EMLR and Pow-TMLR seems to be minor. Pow-EMLR differs from Pow-TMLR only in a simple affine transformation $f_\theta(X) = \frac{1}{\theta}(X - I)$. Note that the composition of affine transformations remains affine, and the FC layer is also an affine transformation. Therefore, Pow-EMLR might be viewed as the approximation of Pow-TMLR. **Based on this discussion, we hypothesize that the tangent classifier serves as the underlying mechanism of matrix functions in GCP.** If this hypothesis holds, Pow-EMLR should perform worse or at least similarly to Pow-TMLR.

To validate this postulation, we conduct experiments on the ImageNet-1k (Deng et al., 2009) and Stanford Cars (Cars) (Krause et al., 2013) datasets. We use the architecture of ResNet-18 and ResNet-50 (He et al., 2016) on the ImageNet and Cars datasets, respectively. Following the classic iSQRT-COV (Li et al., 2018), we set power=$1/2$ and use Newton-Schulz iteration to calculate the matrix square root. Note that Pow-EMLR under Newton-Schulz iteration is exactly the original implementation of iSQRT-COV. As shown in Tab. 3, opposite to our hypothesis, Pow-TMLR is inferior to Pow-EMLR for classifying covariance matrices in GCP. Similar trends are also observed in additional experiments conducted on FGVC datasets, as will be presented in Sec. 6. **These findings suggest that instead of tangent classifiers, there should exist other more fundamental mechanisms for underpinning matrix functions in GCP.**

## 5 MATRIX FUNCTIONS AND RIEMANNIAN CLASSIFIERS

Recently, Riemannian classifiers on the SPD manifold, which can more faithfully respect the innate geometry, have exhibited more promising performance than tangent classifiers (Nguyen & Yang, 2023; Chen et al., 2024a;d). This section will demonstrate that matrix functions in GCP implicitly respect Riemannian classifiers, which offers a unified theoretical explanation of the working mechanism of matrix functions. We start with reviewing the Riemannian SPD classifiers and then present our theoretical analysis in detail.

### 5.1 SPD MULTINOMIAL LOGISTICS REGRESSION REVISITED

Inspired by Lebanon & Lafferty (2004); Ganea et al. (2018), some recent works (Nguyen & Yang, 2023; Chen et al., 2024a;d) extended the Euclidean MLR into SPD manifolds. We first revisit the reformulation of the Euclidean MLR, and then move on to the SPD MLRs introduced by Chen et al. (2024d), especially the ones induced by $(\theta, \alpha, \beta)$-EM and $(\alpha, \beta)$-LEM.

The Euclidean MLR calculates the probability of each class by

$$\forall k \in \{1, \dots, C\}, \quad p(y = k \mid x) \propto \exp\left(\langle a_k, x \rangle - b_k\right), \tag{9}$$

where $x \in \mathbb{R}^n$ is an input vector, $C$ is the number of classes, $b_k \in \mathbb{R}$, and $a_k \in \mathbb{R}^n \backslash \{\mathbf{0}\}$. Eq. (9) can be further rewritten as

$$p(y = k \mid x) \propto \exp\left(\langle a_k, x - p_k \rangle\right), \tag{10}$$

where $p_k$ satisfies $\langle a_k, p_k \rangle = b_k$. As shown in the previous literature (Lebanon & Lafferty, 2004; Ganea et al., 2018), Eq. (10) can be further reformulated by the margin distance to the hyperplane:

$$p(y = k \mid x) \propto \exp(\text{sign}(\langle a_k, x - p_k \rangle) \|a_k\| d(x, H_{a_k, p_k})), \tag{11}$$

where $p_k \in \mathbb{R}^n$ satisfying $\langle a_k, p_k \rangle = b_k$, and the hyperplane $H_{a_k, p_k}$ is defined as:

$$H_{a_k, p_k} = \{x \in \mathbb{R}^n : \langle a_k, x - p_k \rangle = 0\}. \tag{12}$$

Chen et al. (2024d) generalized Eqs. (11) and (12) into general manifolds and proposed the SPD MLRs under five families of metrics. The SPD MLRs under $(\alpha, \beta)$-LEM and $(\theta, \alpha, \beta)$-EM are

$$(\alpha, \beta)\text{-LEM-based: } p(y = k|S) \propto \exp\left[\langle \log(S) - \log(P_k), A_k \rangle^{(\alpha, \beta)}\right], \tag{13}$$

$$(\theta, \alpha, \beta)\text{-EM-based: } p(y = k|S) \propto \exp\left[\frac{1}{\theta}\langle S^\theta - P_k^\theta, A_k \rangle^{(\alpha, \beta)}\right], \tag{14}$$

where $\alpha > 0$, $\beta > -\alpha/n$, and $S$ is an input SPD feature. Here, $P_k \in \mathcal{S}_{++}^n$ and $A_k \in T_I \mathcal{S}_{++}^n \cong \mathcal{S}^n$ are parameters for each class $k$. In Eqs. (13) and (14), the formula within $\exp(\cdot)$ can be viewed as the counterpart of the Euclidean FC layer in SPD manifolds, extracting features to calculate softmax probabilities.

### 5.2 MATRIX FUNCTIONS AS SPD MULTINOMIAL LOGISTICS REGRESSION

Under the standard LEM ($(1, 0)$-LEM) and PEM ($(\theta, 1, 0)$-EM), Eqs. (13) and (14) become

$$\text{LEM-based: } \exp\left[\langle \log(S) - \log(P_k), A_k \rangle\right], \tag{15}$$

$$\text{PEM-based: } \exp\left[\frac{1}{\theta}\langle S^\theta - P_k^\theta, A_k \rangle\right], \tag{16}$$

Eqs. (15) and (16) appear to be far away from the Log-/Pow-EMLR in GCP, as the SPD parameters $\{P_{1\dots C}\}$ requires Riemannian optimization. However, Chen et al. (2024a, Prop. 5.1) show that under the LEM-based Riemannian Stochastic Gradient Descent (RSGD) for each $P_k$ and Euclidean SGD for each $A_k$, Eq. (15) is equivalent to a Euclidean MLR optimized by the Euclidean SGD in the co-domain of the matrix logarithm. Similar to LEM, we have the following proposition w.r.t. PEM.

Table 4: Intrinsic explanations of some classifiers for GCP. For Cho-TMLR, $\tilde{L} = \text{Chol}(S^\theta)$. For Pow-TMLR, $\theta_0 = \frac{\theta_1+\theta_2}{2}$ for $(\theta_1, \theta_2)$-EM, $\theta_0 = \theta$ for $(\theta, \alpha, \beta)$-EM, $\theta_0 = 2\theta$ for $2\theta$-BWM and $(2\theta, \phi_{2\theta}(S))$-BWM. Here, $f_s(\cdot)$ denotes the softmax, $\mathcal{F}(\cdot)$ denotes the FC layer, and $\tilde{V} = \frac{1}{\theta}\left[\lfloor\tilde{L}\rfloor + \lfloor\tilde{L}\rfloor^\top + 2\,\text{Dlog}(\mathbb{D}(\tilde{L}))\right]$ with $S^\theta = \tilde{L}\tilde{L}^\top$ as the Cholesky decomposition.

| | Log-EMLR | Pow-EMLR | ScalePow-EMLR | Pow-TMLR | Cho-TMLR |
|---|---|---|---|---|---|
| Expression | $f_s\left(\mathcal{F}\left(f_{\text{vec}}\left(\text{mlog}(S)\right)\right)\right)$ | $f_s\left(\mathcal{F}\left(f_{\text{vec}}\left(S^\theta\right)\right)\right)$ $(\theta > 0)$ | $f_s\left(\mathcal{F}\left(f_{\text{vec}}\left(\frac{1}{\theta}S^\theta\right)\right)\right)$ $(\theta > 0)$ | $f_s\left(\mathcal{F}\left(f_{\text{vec}}\left(\frac{1}{\theta_0}(S^{\theta_0} - I)\right)\right)\right)$ | $f_s\left(\mathcal{F}\left(f_{\text{vec}}\left(\tilde{V}\right)\right)\right)$ |
| Explanation | SPD MLR | SPD MLR | SPD MLR | Tangent Classifier | Tangent Classifier |
| Metrics | LEM | $(\theta, 1, 0)$-EM | $(\theta, 1, 0)$-EM | $(\theta, \alpha, \beta)$-EM, $(\theta_1, \theta_2)$-EM, $2\theta$-BWM, $(2\theta, \phi_{2\theta}(S))$-BWM | $\theta$-LCM |
| Used in GCP | ✓(Eq. (4)) | ✓ ($\theta = 0.5$ in Eq. (5)) | ✗ | ✗ | ✗ |
| Reference | (Chen et al., 2024a, Prop. 5.1) | Thm. 2 | Thm. 2 | Tab. 2 | Tab. 2 |

**Theorem 2.** [↓] *Under PEM with $\theta > 0$, optimizing each SPD parameter $P_k$ in Eq. (16) by PEM-based RSGD and Euclidean parameter $A_k$ by Euclidean SGD, the PEM-based SPD MLR is equivalent to a Euclidean MLR illustrated in Eq. (10) in the co-domain of $\phi_\theta(\cdot) : \mathcal{S}_{++}^n \to \mathcal{S}_{++}^n$, defined as*

$$\phi_\theta(S) = \frac{1}{\theta}S^\theta, \theta > 0, \forall S \in \mathcal{S}_{++}^n. \tag{17}$$

We define ScalePow-EMLR as $\text{softmax}\left(\mathcal{F}\left(f_{\text{vec}}\left(\frac{1}{\theta}S^\theta\right); A, b\right)\right)$. Then, ScalePow-EMLR respects the SPD MLR under the standard PEM. The only difference between ScalePow-EMLR and Pow-EMLR (Eq. (5)) is the scalar product before vectorization, which is expected to have minor effects on DNNs. Obviously, we have

$$\mathcal{F}\left(f_{\text{vec}}\left(\frac{1}{\theta}S^\theta\right); A, b\right) = \mathcal{F}\left(f_{\text{vec}}\left(S^\theta\right); \tilde{A}, b\right). \tag{18}$$

where $\tilde{A} = \frac{1}{\theta}A$. Therefore, from a forward perspective, ScalePow-EMLR is equivalent to the original Pow-EMLR. Besides, by scaled initialization and learning rate of $A$, ScalePow-EMLR could be completely equivalent to Pow-EMLR during network training. Note that this analysis cannot be transferred into Pow-TMLR. Please refer to App. G for more details.

Therefore, the Pow-EMLR in GCP is implicitly an SPD MLR induced by $(\theta, 1, 0)$-EM. For the widely used matrix square root normalization, it respects the SPD MLR induced by $(1/2, 1, 0)$-EM. We summarize all the findings in Tab. 4. Besides, Thm. 2 can be easily extended into the case of $\theta < 0$. In this case, our work can also offer theoretical insights for the inverse of covariance ($\theta = -1$) proposed by Rahman et al. (2023). More details are presented in App. J.

## 5.3 THEORETICAL INSIGHTS ON THE MATRIX POWER AND LOGARITHM

Previous studies on GCP (Li et al., 2017; Wang et al., 2020a; Song et al., 2021) have *empirically* demonstrated a clear advantage of the matrix power (particularly matrix square root) over matrix logarithm. This subsection offers novel theoretical insights to disentangle the different performance between the matrix logarithm and power in GCP.

As shown by Tab. 4, both matrix logarithm and matrix power implicitly build SPD MLRs. However, the Riemannian metrics they respect are different. Matrix power respects $(\theta, 1, 0)$-EM, while matrix logarithm respects LEM. Both $(\theta, 1, 0)$-EM and LEM share $O(n)$-invariance (Chen et al., 2024d), a powerful property in characterizing covariance matrices. Besides, $(\theta, 1, 0)$-EM is a deformed metric of LEM, interpolating between the standard EM ($\theta = 1$) and LEM ($\theta \to 0$) (Thanwerdas & Pennec, 2022). The standard EM might suffer from a swelling effect for characterizing SPD matrices (Arsigny et al., 2005), while LEM might over-stretch the eigenvalues of SPD matrices due to the computation of matrix logarithm (Song et al., 2021). Consequently, $(\theta, 1, 0)$-EM represents balanced alternatives between the standard LEM and EM. In addition, as shown by Chen et al. (2024d, Tab. 4), $(\theta, 1, 0)$-EM could perform better than LEM regarding SPD MLR. Therefore, the empirical advantages of matrix power over matrix logarithm in the GCP could be attributed to the characteristics of the underlying Riemannian metrics.

Table 5: Results of iSQRT-COV on four datasets with different covariance matrix classifiers. The backbone network on ImageNet is ResNet-18, while the one on the other three FGVC datasets is ResNet-50. Power is set to be $1/2$ for Pow-TMLR, ScalePow-EMLR and Pow-EMLR.

| Classifier | ImageNet-1k | | Aircrafts | | Birds | | Cars | |
|---|---|---|---|---|---|---|---|---|
| | Top-1 Acc (%) | Top-5 Acc (%) | Top-1 Acc (%) | Top-5 Acc (%) | Top-1 Acc (%) | Top-5 Acc (%) | Top-1 Acc (%) | Top-5 Acc (%) |
| Cho-TMLR | N/A | N/A | 78.97 | 91.81 | 48.07 | 72.59 | 51.06 | 74.33 |
| Pow-TMLR | 71.62 | 89.73 | 69.58 | 88.68 | 52.97 | 77.80 | 51.14 | 74.29 |
| ScalePow-EMLR | 72.43 | 90.44 | 71.05 | 89.86 | 63.48 | 84.69 | 80.31 | 94.07 |
| Pow-EMLR | 73 | 90.91 | 72.07 | 89.83 | 63.29 | 84.66 | 80.43 | 94.15 |

Figure 3: The validation top-5 accuracy on the three FGVC datasets for iSQRT-COV with different classifiers using the ResNet-50 backbone.

## 6 EXPERIMENTS

In this section, we validate the following hypothesis based on our previous theoretical analysis.

**(A1)** As Pow-EMLR approximates the tangent classifier Pow-TMLR, the working mechanism of Pow-EMLR is attributed to the **tangent classifier**;

**(A2)** As both Pow-EMLR and Log-EMLR in GCP are equivalent to Riemannian classifiers, the mechanism of matrix normalization should be attributed to **Riemannian classifiers**.

We implement different classifiers for covariance matrix classification, including the original Pow-EMLR, the tangent classifiers Pow-TMLR and Cho-TMLR, and the intrinsic ScalePow-EMLR. We use the Caltech University Birds (Birds) (Welinder et al., 2010), FGVC Aircrafts (Aircrafts) (Maji et al., 2013), Stanford Cars (Cars) (Krause et al., 2013), and ImageNet-1k (Deng et al., 2009) datasets. As the matrix square root is the most effective matrix function in GCP, we set power = $1/2$. In all experiments, we train the network from scratch. More implementation details are in App. H.

### 6.1 MAIN RESULTS

Notably, although ScalePow-EMLR is equivalent to Pow-EMLR under scaled settings, we implement them under the same network settings for a complete comparison. The results on four datasets are shown in Tab. 5. Our main empirical observations are as follows:

**(1). Pow-EMLR>Pow-TMLR.** Pow-EMLR generally outperforms Pow-TMLR, especially on Cars and Birds datasets. Recalling in Tab. 4, the expression of Pow-EMLR differs from Pow-TMLR only in an affine transformation. However, across all four datasets, Pow-EMLR consistently surpasses Pow-TMLR. On the Birds and Cars datasets, Pow-EMLR outperforms Pow-TMLR by a large margin. For example, on the Birds dataset, the top-5 accuracy of Pow-EMLR and Pow-TMLR is 84.66% and 77.80%, respectively, whereas, on the Cars dataset, it is 94.15% and 74.29%.

**(2). Pow-EMLR≈ScalePow-EMLR.** Pow-EMLR shows comparable performance to ScalePow-EMLR. Recalling in Tab. 4, the only difference between Pow-EMLR and ScalePow-EMLR is a scalar product. Moreover, as discussed in Sec. 5.2 this minor difference can be further solved by scaled initialization of the FC layer. Although we use the same initialization for a fair comparison, Pow-EMLR and ScalePow-EMLR show similar performance.

**(3). Pow-EMLR≫Cho-TMLR.** While Cho-TMLR demonstrates the best performance on the Aircrafts datasets, it exhibits the worst performance on the other two FGVC datasets. On the Cars and Birds datasets, Pow-EMLR surpasses Cho-TMLR by a large margin. The unstable performance of

Cho-TMLR might be attributed to the diagonal logarithm, which might overly stretch the diagonal elements of the Cholesky factor.

Based on the above empirical findings, we can reach the following conclusion. **(A1)** is **refuted** by **(1)**. The inferior performance of Pow-TMLR against Pow-EMLR in **(1)** indicates that Pow-EMLR can not be simply viewed as equivalent to the tangent classifier Pow-TMLR. **(A2)** is **validated** by **(2)**. **(2)** validates our theoretical postulation that the effectiveness of matrix power should be attributed to the Riemannian classifier it implicitly constructs.

**Other findings.** In the first and last observations, tangent classifiers are less effective than the Riemannian classifier. Tangent classifiers can distort the innate geometry of the manifold, as the tangent space is only a local linear approximation of the manifold. In contrast, the Riemannian classifier can faithfully respect the geometry of the manifold. Besides, although Log-EMLR coincides with both tangent and Riemannian classifiers, the real underlying mechanism of matrix logarithm should also be attributed to the Riemannian classifier instead of the tangent classifier.

Table 6: Ablations of Pow-EMLR, ScalePow-EMLR, and Pow-TMLR under different settings.

(a) Results of different powers under the ResNet-50.

| Classifier | Aircrafts | | Cars | |
|---|---|---|---|---|
| | Top-1 Acc (%) | Top-5 Acc (%) | Top-1 Acc (%) | Top-5 Acc (%) |
| Pow-TMLR-0.25 | 65.41 | 86.71 | 41.47 | 66.66 |
| ScalePow-EMLR-0.25 | **72.76** | **90.31** | 61.78 | 84.04 |
| Pow-EMLR-0.25 | 71.47 | 90.04 | **62.88** | 84.14 |
| Pow-TMLR-0.5 | 67.9 | 88.75 | 55.01 | 77.95 |
| ScalePow-EMLR-0.5 | 74.29 | 91.12 | 62.42 | 84.82 |
| Pow-EMLR-0.5 | 74.17 | 91.21 | 62.83 | 84.85 |
| Pow-TMLR-0.7 | 65.92 | 87.49 | 50.68 | 74.12 |
| ScalePow-EMLR-0.7 | **74.26** | **91.15** | **64.22** | **83.67** |
| Pow-EMLR-0.7 | 74.17 | 90.49 | 61.41 | 82.39 |

(b) Results under the AlexNet.

| Dataset | Result | Pow-TMLR | Pow-EMLR |
|---|---|---|---|
| Aircrafts | Top-1 Acc (%) | 38.01 | **65.02** |
| | Top-5 Acc (%) | 74.4 | **87.79** |
| Cars | Top-1 Acc (%) | 28.57 | **59.13** |
| | Top-5 Acc (%) | 59.51 | **82.04** |

## 6.2 TRAINING DYNAMICS AND ABLATIONS

**Training dynamics.** Fig. 3 presents the top-5 validation accuracy curves on three FGVC datasets. Pow-EMLR exhibits comparable performance to ScalePow-EMLR throughout the training. Moreover, Pow-EMLR consistently outperforms Pow-TMLR, particularly on the Cars and Birds datasets. This again suggests that the effectiveness of Pow-EMLR should be attributed to the Riemannian MLR rather than the tangent classifier. Furthermore, we note that the decreasing learning rate plays a crucial role in Cho-TMLR. On the Aircrafts dataset, before the 50th epoch, Cho-TMLR exhibits the worst performance among all classifiers. However, after the 50th epoch, when the learning rate reduces, Cho-TMLR surpasses all the other classifiers. Nonetheless, on the remaining two datasets, Cho-TMLR remains inferior throughout the training. This discrepancy may be attributed to the logarithm operation in Cho-TMLR. Recalling Eq. (8), there is a diagonal logarithm for the Cholesky factor. Similar to the matrix logarithm, Eq. (8) will also over-stretch the diagonal elements of the Cholesky factor, compromising the overall performance of Cho-TMLR.

**Ablations.** To further validate our postulation, we compare Pow-EMLR, SaclePow-EMLR, and Pow-TMLR with different powers under the ResNet-50 architecture, *i.e.,*, $\theta = 0.25, 0.5, 0.7$. We also compare Pow-EMLR against Pow-TMLR under the AlexNet architecture. The ablations are conducted on the Aircrafts and Car datasets. The results discussed below confirm again our findings that the mechanism of matrix functions in GCP should be attributed to Riemannian classifiers.

*Impact of matrix power.* Following Song et al. (2021), we use accurate SVD to calculate the matrix power and Padé approximant for backpropagation. The results are reported in Tab. 6a. Since we use SVD for the matrix power here, the results in Tab. 6a under $\theta = 0.5$ are slightly different from Tab. 5. Nevertheless, Pow-EMLR consistently shows similar performance to ScalePow-EMLR and outperforms Pow-TMLR under different powers.

*Impact of architectures.* We also use the vanilla AlexNet (Krizhevsky et al., 2012) as an alternative backbone. Tab. 6b presents the comparison results under the AlexNet architecture. Consistent with our previous observation, Pow-EMLR still outperforms Pow-TMLR.

## 7 CONCLUSIONS

This paper presents a unified understanding of the role of matrix functions in GCP, including matrix power and logarithm. Our study reveals that matrix functions implicitly construct Riemannian classifiers for classifying covariance matrices, thus justifying the application of the Euclidean classifier after matrix power. We validate our findings through experiments conducted on three FGVC and the large-scale ImageNet datasets. To the best of our knowledge, our work is the **first** to explain the theoretical mechanism behind matrix functions from the perspective of Riemannian geometry. Therefore, our work opens a novel possibility for designing GCP classifiers from a Riemannian perspective. As a future avenue, we will design effective GCP classifiers based on other Riemannian MLRs (Nguyen & Yang, 2023; Chen et al., 2024d).

ACKNOWLEDGMENTS

This work was partly supported by the MUR PNRR project FAIR (PE00000013) funded by the NextGenerationEU, the EU Horizon project ELIAS (No. 101120237), and a donation from Cisco. The authors also gratefully acknowledge the financial support from the China Scholarship Council (CSC), as well as the CINECA award under the ISCRA initiative for the availability of partial HPC resources support.

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

APPENDIX CONTENTS

## A  FUTURE WORK

While Chen et al. (2024d) also explored Riemannian MLRs induced by other metrics, these MLRs involve computationally expensive Riemannian computations, rendering them unsuitable for large-scale datasets. As a future avenue, we aim to simplify the Riemannian computations in these alternative Riemannian classifiers and apply them to GCP for improved covariance matrix classification.

## B  RELATED WORK

**Global covariance pooling.** GCP aims to leverage the second-order statistics of deep features to enhance the learning competence of DNNs. $\text{DeepO}^2\text{P}$ (Ionescu et al., 2015), acknowledged as the first end-to-end global covariance pooling network, employs matrix logarithm for the classification of covariance matrices. This method also offers matrix backpropagation to differentiate the gradient w.r.t the decomposition-based matrix functions. Following this pioneering work, B-CNN (Lin et al., 2015) employs the outer product of global features and carries out element-wise power normalization. However, there exist three limitations of the above two methods. Firstly, the high dimensional covariance feature considerably escalates the parameters of the FC layer, thereby introducing the risk of overfitting. Secondly, the matrix logarithm could over-stretch the small eigenvalues, undermining the effectiveness of GCP. Thirdly, the matrix logarithm is based on matrix decomposition, which is computationally expensive. The subsequent research primarily focuses on four aspects: (a) adopting richer statistical representation (Wang et al., 2017; Zheng et al., 2019; Nguyen, 2021); (b) reducing the dimensionality of the covariance feature (Gao et al., 2016; Kong & Fowlkes, 2017; Cui et al., 2017; Acharya et al., 2018; Rahman et al., 2020; Wang et al., 2022a); (c) investigating effective and efficient matrix normalization (Li et al., 2018; Zheng et al., 2019; Lin & Maji, 2017; Yu et al., 2020; Song et al., 2022c;b); (d) improving covariance conditioning for better generalization ability (Song et al., 2022d;a). In this work, we do not aim to achieve state-of-the-art performance over the existing GCP-based methods but rather to unravel the underlying theoretical mechanism of GCP matrix functions.

**Interpretations of global covariance pooling.** Along with the progress of GCP, several works began to study its mechanism. Wang et al. (2020b) investigated the effect of GCP on deep Convolutional Neural Networks (CNNs) from an optimization perspective, including accelerated convergence, stronger robustness, and good generalization ability. Wang et al. (2023) further broadened these investigations, substantiating the merits of GCP in other networks, such as vision transformers (Touvron et al., 2021; Yuan et al., 2021; Liu et al., 2021) and differentiable Neural Architecture Search (NAS) (Liu et al., 2019). Song et al. (2021) empirically studied the advantage of approximate matrix square root over the accurate one. Wang et al. (2022a) considered the matrix power as decorrelating representations and developed a channel-adaptive dropout to produce lower-dimensional covariance matrices. Nevertheless, existing literature does not fully address the fundamental question of why Euclidean classifiers operate effectively in the non-Euclidean space generated by the matrix power. Our research fills in this theoretical gap, offering intrinsic explanations regarding the role of the matrix functions in GCP.

**Riemannian classifiers on SPD manifolds.** Since the matrix logarithm is a diffeomorphism between the SPD manifold and its tangent space at the identity (Arsigny et al., 2005), the most widely used classifier on SPD manifolds is composed of the matrix logarithm and a Euclidean classifier (Wang et al., 2021; Chen et al., 2023b; Wang et al., 2022b; Nguyen, 2022a;b; Wang et al., 2022c; Chen et al., 2024b; Wang et al., 2024b). However, this tangent classifier might distort the intrinsic geometry of SPD manifolds. Similar issues also arise in other manifolds (Huang et al., 2017; Wang et al., 2024a; Chen et al., 2025) Inspired by HNNs (Ganea et al., 2018), recent studies have developed intrinsic classifiers directly on SPD manifolds. Nguyen & Yang (2023) introduced three gyro structures on SPD manifolds induced by AIM, LEM, and LCM, respectively. Based on these gyro structures, the authors generalize the Euclidean Multinomial Logistics Regression (MLR). Concurrently, Chen et al. (2024a) proposed a formula for SPD MLR under Riemannian metrics pulled back from the Euclidean space. However, both works require specific Riemannian properties and focus on certain metrics on SPD manifolds. Chen et al. (2024d) presented a general framework for designing Riemannian MLRs on general geometries and showcased their framework under various metrics on SPD manifolds, covering the SPD MLRs introduced by Chen et al. (2024a); Nguyen & Yang (2023). Based on this framework, Chen et al. (2024c) showcased the SPD MLR under their

Table 7: Summary of notations.

| Notation | Explanation |
| --- | --- |
| $\mathcal{S}_{++}^n$ | The SPD manifold |
| $\mathcal{S}^n$ | The Euclidean space of symmetric matrices |
| $\mathcal{L}^n$ | The Euclidean space of $n \times n$ lower triangular matrices |
| $T_P \mathcal{S}_{++}^n$ | The tangent space at $P \in \mathcal{S}_{++}^n$ |
| $g_P(\cdot, \cdot)$ or $\langle \cdot, \cdot \rangle_P$ | The Riemannian metric at $P \in \mathcal{S}_{++}^n$ |
| $\langle \cdot, \cdot \rangle$ or $\cdot : \cdot$ | The standard Frobenius inner product |
| $\mathrm{Log}_P$ | The Riemannian logarithm at $P$ |
| $H_{a,p}$ | The Euclidean hyperplane |
| $f_{*,P}$ | The differential map of $f$ at $P \in \mathcal{S}_{++}^n$ |
| $f^* g$ | The pullback metric by $f$ from $g$ |
| $\mathrm{ad}(\cdot)$ | The adjoint operator of linear maps |
| **ST** | $\mathbf{ST} = \{(\alpha, \beta) \in \mathbb{R}^2 \mid \min(\alpha, \alpha + n\beta) > 0\}$ |
| $\langle \cdot, \cdot \rangle^{(\alpha, \beta)}$ | The O$(n)$-invariant Euclidean inner product |
| $g^{(\alpha, \beta)\text{-LE}}$ | The Riemannian metric of $(\alpha, \beta)$-LEM |
| $g^{(\alpha, \beta)\text{-AI}}$ | The Riemannian metric of $(\alpha, \beta)$-AIM |
| $g^{(\theta, \alpha, \beta)\text{-AI}}$ | The Riemannian metric of $(\theta, \alpha, \beta)$-AIM |
| $g^{(\alpha, \beta)\text{-E}}$ | The Riemannian metric of $(\alpha, \beta)$-EM |
| $g^{(\theta, \alpha, \beta)\text{-E}}$ | The Riemannian metric of $(\theta, \alpha, \beta)$-EM |
| $g^{(\theta_1, \theta_2)\text{-E}}$ | The Riemannian metric of $(\theta_1, \theta_2)$-EM |
| $g^{\mathrm{BW}}$ | The Riemannian metric of BWM |
| $g^{M\text{-BW}}$ | The Riemannian metric of $M$-BWM |
| $g^{(2\theta, M)\text{-BW}}$ | The Riemannian metric of $(2\theta, M)$-BWM |
| $g^{\mathrm{LC}}$ | The Riemannian metric of LCM |
| $g^{\theta\text{-LC}}$ | The Riemannian metric of $\theta$-LCM |
| $f_{\mathrm{FC}}$ or $\mathcal{F}(\cdot; A, b)$ | The FC layer |
| $\mathrm{Pow}_\theta$ or $(\cdot)^\theta$ | The matrix power |
| $f_{\mathrm{vec}}$ | The vectorization |
| $f_{\mathrm{EC}}$ | A Euclidean classifier |
| mlog | The matrix logarithm |
| $\mathcal{L}_P[\cdot]$ | The Lyapunov operator |
| Chol | The Cholesky decomposition |
| $\mathcal{L}_{P,M}[\cdot]$ | The generalized Lyapunov operator |
| $\mathrm{Dlog}(\cdot)$ | The diagonal element-wise logarithm |
| $f_{\mathrm{M}}$ | The matrix function of matrix power or logarithm |
| $\lfloor \cdot \rfloor$ | The strictly lower triangular part of a square matrix |
| $\mathbb{D}(\cdot)$ | A diagonal matrix with diagonal elements from a square matrix |

proposed product metrics. In this paper, based on the Riemannian classifiers developed by Chen et al. (2024d), we present an intrinsic explanation for matrix functions in GCP.

## C  NOTATIONS AND ABBREVIATIONS

For better clarity, we summarize all the notations in Tab. 7 and all the abbreviations in Tab. 8.

## D  ADDITIONAL PRELIMINARIES

### D.1  PULLBACK METRICS

The power-deformed metrics on the SPD manifold are special cases of pullback metrics. Pullback metrics are common techniques in Riemannian geometry, connecting different Riemannian metrics.

Table 8: Summary of Abbreviations.

| Abbreviation | Explanation |
|---|---|
| SPD | Symmetric Positive Definite |
| GCP | Global covariance pooling |
| GAP | Global Average Pooling |
| LEM | Log-Euclidean Metric |
| AIM | Affine-Invariant Metric |
| EM | Euclidean Metric |
| PEM | Power Euclidean Metric |
| MPEM | Mixed Power Euclidean Metric |
| BWM | Bures-Wasserstein Metric |
| GBWM | Generalized Bures-Wasserstein Metric |
| FGVC | Fine-Grained Visual Categorization |
| MLR | Multinomial Logistics Regression |
| EMLR | Euclidean Multinomial Logistics Regression |
| RMLR | Riemannian Multinomial Logistics Regression |
| SPD MLR | RMLR on SPD manifolds |
| Log-EMLR | Eq. (4) |
| Pow-EMLR | Eq. (5) |
| Pow-TMLR | EMLR in the tangent space generated by Eq. (7) |
| ScalePow-EMLR | ScalePow-EMLR in Tab. 4 |
| Cho-TMLR | EMLR in the tangent space generated by Eq. (8) |

**Definition 3** (Pullback Metrics). Suppose $\mathcal{M}, \mathcal{N}$ are smooth manifolds, $g$ is a Riemannian metric on $\mathcal{N}$, and $f : \mathcal{M} \to \mathcal{N}$ is smooth. Then the pullback of $g$ by $f$ is defined point-wisely,

$$(f^*g)_p(V_1, V_2) = g_{f(p)}(f_{*,p}(V_1), f_{*,p}(V_2)), \tag{19}$$

where $p \in \mathcal{M}$, $f_{*,p}(\cdot)$ is the differential map of $f$ at $p$, and $V_i \in T_p\mathcal{M}$. If $f^*g$ is positive definite, it is a Riemannian metric on $\mathcal{M}$, which is called the pullback metric defined by $f$.

### D.2 RIEMANNIAN OPERATORS ON THE SPD MANIFOLD

The $O(n)$-invariant Euclidean inner product on $\mathcal{S}^n$ (Thanwerdas & Pennec, 2023) is defined as

$$\langle V, W \rangle^{(\alpha,\beta)} = \alpha\langle V, W \rangle + \beta\operatorname{tr}(V)\operatorname{tr}(W), \tag{20}$$

where $(\alpha, \beta) \in \mathbf{ST}$ with $\mathbf{ST} = \{(\alpha, \beta) \in \mathbb{R}^2 \mid \min(\alpha, \alpha + n\beta) > 0\}$, $V, W \in \mathcal{S}^n$, and $\langle \cdot, \cdot \rangle$ is the standard matrix inner product.

We summarize deformed SPD metrics and associated Riemannian operators in Tab. 9 with the following notations. Specifically, $P, Q, M \in \mathcal{S}_{++}^n$ are SPD matrices, and $V, W$ are tangent vectors in the tangent space at $P$, *i.e.,* $T_P\mathcal{S}_{++}^n$. We denote $g_P(\cdot, \cdot)$ as the Riemannian metric at $P$, and $\operatorname{Log}_P(\cdot)$ as the Riemannian logarithm at $P$, respectively. Also, $\operatorname{Chol}$ and $\operatorname{mlog}$ represent the Cholesky decomposition and matrix logarithm, with their differential maps at $P$ denoted as $\operatorname{Chol}_{*,P}$ and $\operatorname{mlog}_{*,P}$, respectively. We denote $\tilde{V} = \operatorname{Chol}_{*,P}(V)$, $\tilde{W} = \operatorname{Chol}_{*,P}(W)$, $L = \operatorname{Chol}(P)$, and $K = \operatorname{Chol}(Q)$. $\lfloor \cdot \rfloor$ is the strictly lower part of a square matrix, $\mathbb{D}(\cdot)$ is a diagonal matrix with diagonal elements of a square matrix, and $\operatorname{Dlog}(\cdot)$ is a diagonal matrix consisting of the logarithm of the diagonal entries of a square matrix. We denote $\mathcal{L}_{P,M}[V]$ as the generalized Lyapunov operator, *i.e.,* the solution to the matrix linear system $M\mathcal{L}_{P,M}[V]P + P\mathcal{L}_{P,M}[V]M = V$. When $M = I$, $\mathcal{L}_{P,I}[V]$ is reduced to the Lyapunov operator, denoted as $\mathcal{L}_P[V]$.

## E TECHNICAL DETAILS ON RIEMANNIAN LOGARITHM

We first review a well-known result for the pullback metric (Thanwerdas & Pennec, 2022, Tab. 2).

Table 9: Riemannian operators and deformed metrics of seven basic metrics on SPD manifolds. Note that for MPEM, $P$ and $Q$ must be commuting matrices when computing the Riemannian logarithm.

| Name | Riemannian Metric $g_P(V,W)$ | Riemannian Logarithm $\mathrm{Log}_P Q$ | Deformation $(\theta \neq 0)$ |
|---|---|---|---|
| $(\alpha,\beta)$-LEM (Thanwerdas & Pennec, 2023) | $\langle \mathrm{mlog}_{*,P}(V), \mathrm{mlog}_{*,P}(W) \rangle^{(\alpha,\beta)}$ | $(\mathrm{mlog}_{*,P})^{-1} [\mathrm{mlog}(Q) - \mathrm{mlog}(P)]$ | $\frac{1}{\theta^2} \mathrm{Pow}_\theta^* g^{(\alpha,\beta)\text{-LE}}$ |
| $(\alpha,\beta)$-AIM (Thanwerdas & Pennec, 2023) | $\langle P^{-1}V, WP^{-1} \rangle^{(\alpha,\beta)}$ | $P^{1/2} \mathrm{mlog}\left(P^{-1/2}QP^{-1/2}\right) P^{1/2}$ | $\frac{1}{\theta^2} \mathrm{Pow}_\theta^* g^{(\alpha,\beta)\text{-AI}}$ |
| $(\alpha,\beta)$-EM (Thanwerdas & Pennec, 2023) | $\langle V, W \rangle^{(\alpha,\beta)}$ | $Q - P$ | $\frac{1}{\theta^2} \mathrm{Pow}_\theta^* g^{(\alpha,\beta)\text{-E}}$ |
| $(\theta_1,\theta_2)$-EM (Thanwerdas & Pennec, 2022) | $\frac{1}{\theta_1\theta_2}\langle \mathrm{Pow}_{\theta_1*,P}(V), \mathrm{Pow}_{\theta_2*,P}(W) \rangle$ | $(\mathrm{Pow}_{\theta*,P})^{-1}(Q^\theta - P^\theta)$, with $\theta = (\theta_1 + \theta_2)/2$ | N/A |
| LCM (Lin, 2019) | $\sum_{i>j} \tilde{V}_{ij}\tilde{W}_{ij} + \sum_{j=1}^n \tilde{V}_{jj}\tilde{W}_{jj}L_{jj}^{-2}$ | $(\mathrm{Chol}^{-1})_{*,L}\left[\lfloor K \rfloor - \lfloor L \rfloor + \mathbb{D}(L)\,\mathrm{Dlog}(\mathbb{D}(L)^{-1}\mathbb{D}(K))\right]$ | $\frac{1}{\theta^2} \mathrm{Pow}_\theta^* g^{\mathrm{LC}}$ |
| BWM (Bhatia et al., 2019) | $\frac{1}{2}\langle \mathcal{L}_P[V], W \rangle$ | $(PQ)^{1/2} + (QP)^{1/2} - 2P$ | $\frac{1}{4\theta^2} \mathrm{Pow}_{2\theta}^* g^{\mathrm{BW}}$ |
| GBWM (Han et al., 2023) | $\frac{1}{2}\langle \mathcal{L}_{P,M}[V], W \rangle$ | $M\left(M^{-1}PM^{-1}Q\right)^{1/2} + \left(QM^{-1}PM^{-1}\right)^{1/2} M - 2P$ | $\frac{1}{4\theta^2} \mathrm{Pow}_{2\theta}^* g^{M\text{-BW}}$ |

**Lemma 4.** *Given a Riemannian metric $g$ on the SPD manifold $\mathcal{S}_{++}^n$ and a diffeomorphism $f: \mathcal{S}_{++}^n \to \mathcal{S}_{++}^n$, the Riemannian logarithm $\tilde{\mathrm{Log}}_P$ under the pullback metric $\tilde{g} = f^*g$ is*

$$\tilde{\mathrm{Log}}_P Q = (f_{*,P})^{-1}\left(\mathrm{Log}_{f(P)} f(Q)\right), \tag{21}$$

*where $f_{*,P}$ is the differential map at $P$, and $\mathrm{Log}$ is the Riemannian logarithm under $g$.*

Next, we show a lemma about the scaling of a Riemannian metric.

**Lemma 5.** *Supposing $\mathcal{S}_{++}^n$ is endowed with a Riemannian metric $g$ and $a > 0$ is a positive real scalar, the scaling metric $ag$ shares the same Riemannian logarithm map with $g$.*

*Proof.* Since the Christoffel symbols of $ag$ are identical to those of $g$, the geodesic functions under both $ag$ and $g$ remain unchanged. This implies that the Riemannian exponential maps are the same for $ag$ and $g$. As the inverse of the Riemannian exponential maps, the Riemannian logarithm maps under $ag$ and $g$ are also identical. □

By the above lemmas, we can readily prove Tab. 2.

*Proof.* By Lem. 5, for the power-deformed metric of a metric $g$ in $\mathcal{S}_{++}^n$, the Riemannian logarithm at $I$ is the same as the counterpart under $\mathrm{Pow}_\theta^* g$. Therefore, in the following, without loss of generality, we compute $\mathrm{Log}_I$ under $\mathrm{Pow}_\theta^* g$. We further denote the Riemannian logarithm under $g$ as $\tilde{\mathrm{Log}}$.

In the following, we denote $P$ as an SPD matrix, $0$ as the $n \times n$ zero matrix, and $V$ as a tangent vector in $T_I \mathcal{S}_{++}^n$. Besides, we note that

$$\mathrm{Pow}_{\theta*,I}(V) = \theta V. \tag{22}$$

We first deal with $(\alpha,\beta)$-LEM and $\theta$-LCM, as both of them are pullback metrics from the Euclidean space. Then, we proceed to deal with other metrics

$(\alpha,\beta)$**-LEM:** As shown by Thanwerdas & Pennec (2023), the Riemannian logarithm at $I$ is

$$\begin{aligned}
\mathrm{Log}_I(P) &= \mathrm{mlog}_{*,I}^{-1}\left(\mathrm{mlog}(P) - \mathrm{mlog}(I)\right) \\
&= \mathrm{mlog}(P).
\end{aligned} \tag{23}$$

$\theta$**-LCM:** We define a map as

$$f = \psi \circ \mathrm{Chol} \circ \mathrm{Pow}_\theta, \tag{24}$$

where $\psi(L) = \lfloor L \rfloor + \mathrm{Dlog}(\mathbb{D}(L))$ for the lower triangular matrix $L$. Chen et al. (2024e) shows that LCM is the pullback metric by $\psi \circ \mathrm{Chol}$ from the Euclidean space $\mathcal{L}^n$ of lower triangular matrices.

Therefore, $\text{Pow}_\theta^* g^{\text{LC}}$ is the pullback metric from $\mathcal{L}^n$ by $f$. Besides, we have the following:

$$f(P) = \lfloor \tilde{L} \rfloor + \text{Dlog}(\mathbb{D}(\tilde{L})), \tag{25}$$

$$f(I) = 0, \tag{26}$$

$$f_{*,I}(V) = \theta \left( \lfloor V \rfloor + \frac{1}{2} \mathbb{D}(L) \right), \tag{27}$$

where $\tilde{L} = \text{Chol}(P^\theta)$. We have

$$\begin{aligned}
\text{Log}_I(P) &= (f_{*,P})^{-1}(f(P) - f(I)) \\
&= \frac{1}{\theta} \left[ \lfloor \tilde{L} \rfloor + \lfloor \tilde{L} \rfloor^\top + 2 \, \text{Dlog}(\mathbb{D}(\tilde{L})) \right].
\end{aligned} \tag{28}$$

For $(\theta, \alpha, \beta)$-EM, $(\theta_1, \theta_2)$-EM, $(\theta, \alpha, \beta)$-AIM, $2\theta$-BWM, and $(2\theta, P^{2\theta})$-BWM, we denote $\text{Log}_I$ as their logarithm at $I$, while $\bar{\text{Log}}_I$ as the logarithm under the metric before deformation. The results can be directly obtained by Eq. (22), Lem. 4, Lem. 5, and Tab. 9.

$(\theta, \alpha, \beta)$-**EM:**

$$\begin{aligned}
\text{Log}_I(P) &= \frac{1}{\theta} \bar{\text{Log}}_I(P^\theta) \\
&= \frac{1}{\theta} \left( P^\theta - I \right).
\end{aligned} \tag{29}$$

$(\theta_1, \theta_2)$-**EM:** The $\bar{\text{Log}}_I$ can be directly obtained by Tab. 9 and Eq. (22).

$(\theta, \alpha, \beta)$-**AIM:**

$$\begin{aligned}
\text{Log}_I(P) &= \frac{1}{\theta} \bar{\text{Log}}_I(P^\theta) \\
&= \frac{1}{\theta} \, \text{mlog}(P^\theta) \\
&= \text{mlog}(P).
\end{aligned} \tag{30}$$

$2\theta$-**BWM:**

$$\begin{aligned}
\text{Log}_I(P) &= \frac{1}{2\theta} \bar{\text{Log}}_I(P^{2\theta}) \\
&= \frac{1}{\theta} (P^\theta - I).
\end{aligned} \tag{31}$$

$(2\theta, P^{2\theta})$-**BWM:** Under $M$-BWM, we have

$$\text{Log}_I(M) = 2(M^{\frac{1}{2}} - I). \tag{32}$$

Therefore, for $(2\theta, P^{2\theta})$-BWM, we have

$$\begin{aligned}
\text{Log}_I(P) &= \frac{1}{2\theta} \bar{\text{Log}}_I(P^{2\theta}) \\
&= \frac{1}{\theta} (P^\theta - I).
\end{aligned} \tag{33}$$

$\square$

## F  POWER-DEFORMED GBWM AS LOCAL POWER-AIM

Let us first formalize this property.

**Proposition 6.** *For any $P \in \mathcal{S}_{++}^n$ and $V, W \in T_P \mathcal{S}_{++}^n$, we have the following:*

$$g_P^{(2\theta, P^{2\theta})\text{-}BW}(V, W) = \frac{1}{4} g_P^{(2\theta, 1, 0)\text{-}AI}(V, W). \tag{34}$$

*Proof.* As shown by Bhatia (2009), the Riemannian metric of the standard AIM (($(1, 1, 0)$-AIM) is

$$g_P^{\text{AI}}(V, W) = \text{vec}(V)^\top (P \otimes P)^{-1} \text{vec}(W), \tag{35}$$

where $\text{vec}(V)$ is the column vectorization of $V$, $\otimes$ is the Kronecker product.

For the $(2\theta, P^{2\theta})$-BWM, we have the following:

$$
\begin{aligned}
g_P^{(2\theta, P^{2\theta})\text{-BW}}(V, W) &= \frac{1}{4\theta^2} g_{\tilde{P}}^{\phi_{2\theta}(P)\text{-BW}}(\tilde{V}, \tilde{W}) \\
&= \frac{1}{4} \cdot \frac{1}{4\theta^2} \text{vec}(\tilde{V})^\top (\tilde{P} \otimes \tilde{P})^{-1} \text{vec}(\tilde{W}) \\
&= \frac{1}{4} \cdot \frac{1}{4\theta^2} g_{\tilde{P}}^{\text{AI}}(\tilde{V}, \tilde{W}) \\
&= \frac{1}{4} g_P^{(2\theta, 1, 0)\text{-AI}}(V, W),
\end{aligned}
\tag{36}
$$

where $\tilde{V} = \text{Pow}_{2\theta*, P}(V)$, $\tilde{W} = \text{Pow}_{2\theta*, P}(W)$, $\tilde{P} = P^{2\theta}$, and Eq. (36) can be obtain by Han et al. (2023, Eq. 3) □

## G ADDITIONAL DISCUSSIONS ON POW-TMLR, POW-EMLR, AND SCALEPOW-EMLR

### G.1 THE EQUIVALENCE OF POW-EMLR AND SCALEPOW-EMLR

It can be proven that Pow-EMLR is equivalent to ScalePow-EMLR under scaled initial weight and learning rate in the FC layer. We denote the network as

$$x_0 \in \mathbb{R}^{d_0} \xrightarrow{g(\cdot; \Theta)} x \in \mathbb{R}^d \xrightarrow{f_{\text{FC}}} y \in \mathbb{R}^c \to L \in \mathbb{R}, \tag{37}$$

where $x_0$, $g(\cdot; \Theta)$, $f_{\text{FC}}$, and $L$ are the input feature, feature extraction with parameter $\Theta$, FC layer, and loss, respectively. The FC layers in Pow-EMLR and ScalePow-EMLR are denoted as $y = Ax + b$ and $\bar{y} = \frac{1}{\theta} \bar{A}\bar{x} + \bar{b}$. We set the initial values and learning rates of $A$ and $\bar{A}$ satisfying $A_0 = \frac{1}{\theta}\bar{A}_0$ and $\bar{\gamma} = \theta^2 \gamma$, and maintain all the other settings the same. Then, we have the following for the gradient at $A = A_0$ (or $\bar{A} = \bar{A}_0$):

$$
\begin{aligned}
\frac{\partial L}{\partial \bar{A}} &= \frac{1}{\theta} \frac{\partial L}{\partial \bar{y}} \bar{x}^\top = \frac{1}{\theta} \frac{\partial L}{\partial y} x^\top = \frac{1}{\theta} \frac{\partial L}{\partial A}, \\
\frac{\partial L}{\partial \bar{x}} &= \frac{1}{\theta} \bar{A}^\top \frac{\partial L}{\partial \bar{y}} = A^\top \frac{\partial L}{\partial y} = \frac{\partial L}{\partial x}.
\end{aligned}
\tag{38}
$$

Under SGD, the updated values of $\bar{A}$ satisfying the following:

$$
\begin{aligned}
\frac{1}{\theta}\bar{A}_1 &= \frac{1}{\theta}(\bar{A}_0 + \bar{\gamma}\frac{\partial L}{\partial \bar{A}}) \\
&= \frac{1}{\theta}(\bar{A}_0 + \bar{\gamma}\frac{1}{\theta}\frac{\partial L}{\partial A}) \\
&= \frac{1}{\theta}\bar{A}_0 + \bar{\gamma}\frac{1}{\theta^2}\frac{\partial L}{\partial A} \\
&= A_0 + \gamma\frac{\partial L}{\partial A} \\
&= A_1.
\end{aligned}
\tag{39}
$$

Therefore, the updated values of $A$ and $\bar{A}$ still satisfy $A_1 = \frac{1}{\theta}\bar{A}_1$. In addition, the gradients of Pow-EMLR w.r.t. $x$ and $b$ are identical to ScalePow-EMLR w.r.t. $\bar{x}$ and $\bar{b}$. Therefore, Pow-EMLR is equivalent to ScalePow-EMLR under scaled settings.

## G.2 THE IN-EQUIVALENCE OF POW-EMLR AND POW-TMLR

We denote $X = S^\theta$. Then for Pow-TMLR, we have the following

$$
\begin{aligned}
y &= \mathcal{F}\left(f_{\text{vec}}\left(\frac{1}{\theta}(X + I)\right); A, b\right) \\
&= \mathcal{F}\left(f_{\text{vec}}(X + I); \tilde{A}, b\right) \\
&= \mathcal{F}\left(f_{\text{vec}}(X); \tilde{A}, \tilde{b}\right),
\end{aligned}
\tag{40}
$$

where $\tilde{A} = \frac{1}{\theta}A$ and $\tilde{b} = \frac{1}{\theta}A f_{\text{vec}}(I)$.

As $A$ appears in $\tilde{b}$, the gradient of $A$ is composed of two parts, one w.r.t. $y$ and the other one w.r.t. $\tilde{b}$. In contrast, in the standard FC layer $y = \mathcal{F}(X; A, b)$, the gradient of $A$ is independent of $b$. Therefore, Pow-TMLR cannot be simply viewed as equivalent to Pow-EMLR with transformed initialization.

## G.3 A RIEMANNIAN PERSPECTIVE OF POW-TMLR VS. POW-EMLR

Although the numerical expressions of Pow-TMLR and Pow-EMLR differ by a constant transformation, they differ fundamentally in theory: Pow-TMLR is a tangent classifier, whereas Pow-EMLR is a Riemannian classifier.

1. **Tangent Classifier:** The tangent classifier treats the entire manifold as a single tangent space at the identity matrix. When mapping data into this tangent space, critical structural information, such as distances and angles, cannot be preserved. This distortion undermines classification performance. In contrast, Riemannian MLR is constructed based on Riemannian geometry, fully respecting the manifold's geometric structure.

2. **Tangent as a Special Case of Riemannian Classifier**. The tangent classifier can be seen as a reduced case of the Riemannian classifier. For example, let us take Eq. (16) as an example. When all SPD parameters $P_k$ are set to the fixed identity matrix, Eq. (16) exactly corresponds to Pow-TMLR.

In summary, the Riemannian classifier enjoys significant theoretical advantages over the tangent classifier while incorporating the tangent classifier as a special case.

# H ADDITIONAL EXPERIMENTAL DETAILS

## H.1 DATASETS

The Caltech University Birds (Birds) (Welinder et al., 2010) dataset is composed of $11,788$ images distributed over 200 different bird species. The FGVC Aircrafts (Aircrafts) (Maji et al., 2013) dataset comprises $10,000$ images of 100 classes of airplanes, while the Stanford Cars (Cars) (Krause et al., 2013) dataset consists of $16,185$ images representing 196 classes of cars. In addition to these widely used FGVC datasets, we also evaluate our proposed theory on the large-scale ImageNet-1k (Deng et al., 2009) dataset, which contains 1.28M training images, 50K validation images and 100K testing images distributed across 1K classes.

## H.2 IMPLEMENTATION DETAILS

We follow the official Pytorch code of iSQRT-COV[1] (Li et al., 2018) to reimplement GCP. Following Wang et al. (2020a); Song et al. (2022a), we use ResNet-18 as our backbone network on the ImageNet dataset, and ResNet-50 on the other three FGVC datasets. On Both the ImageNet-1k and FGVC datasets, the ResNet-18 and ResNet-50 are trained from scratch with the GCP layer.

---

[1]https://github.com/jiangtaoxie/fast-MPN-COV

As the matrix square root is the most effective matrix function in GCP, we set power = $1/2$ for matrix power normalization. Following Song et al. (2022a), we reduce the channels of the final convolutional features from 2048 to 256 for compact representation of covariance matrices, producing $256 \times 256$ spatial covariance matrices. We train the network from scratch with an SGD optimizer on all datasets. For a fair comparison, the learning settings are identical for Pow-EMLR and ScalePow-EMLR, while we fine-tune Pow-TMLR w.r.t. learning rate, classifier factor, and batch size on three FGVC datasets. The learning rate of the FC layer is set to be $k$ times larger than the convolutional layers, where $k$ is called the classifier factor. We use a step-wise learning scheduler, dividing the learning rate by 5 at $n$-th epoch. Tab. 10 summarizes the hyperparameters in our main experiments in Tab. 5.

For Cho-TMLR, the learning rate is set as $1e^{-2}$, $5e^{-3}$, and $3e^{-3}$ for the convolutional layers on the Aircrafts, Birds, and Cars dataset. The batch size on the Cars dataset is 4. On the Aircrafts dataset, the training lasts 60 epochs with the learning rate divided by 5 at epoch 50. On the Cars and Birds datasets, the training lasts 120 epochs with a learning rate reduction by a divisor of 10 at epoch 100.

The experiments on ImageNet use a workstation with 32-core AMD EPYC 7302 CPU and an NVIDIA RTX A6000, while other experiments use a workstation with 16-core AMD EPYC 7302 CPU and an NVIDIA GeForce RTX 2080 Ti GPU. Due to the heavy computational burden of Cholesky decomposition, we do not implement Cho-TMLR on the ImageNet.

Table 10: Summary of hyperparameters.

| Backbone | ResNet18 | ResNet50 | | | | |
|---|---|---|---|---|---|---|
| Dataset | ImageNet | Aircrafts | Birds | | Cars | |
| Classifiers | Pow-EMLR ScalePow-EMLR Pow-TMLR | Pow-EMLR ScalePow-EMLR Pow-TMLR | Pow-EMLR ScalePow-EMLR | Pow-TMLR | Pow-EMLR ScalePow-EMLR | Pow-TMLR |
| Learning Rate | $1e^{-1.1}$ | $5e^{-3}$ | $5e^{-2}$ | $5e^{-3}$ | $5e^{-2}$ | $5e^{-3}$ |
| Weight Decay | $1e^{-4}$ | $1e^{-4}$ | $1e^{-4}$ | $1e^{-4}$ | $1e^{-4}$ | $1e^{-4}$ |
| Classifier Factor | 1 | 5 | 10 | 1 | 10 | 1 |
| LR Scheduler | $[30, 45, 60]$ | $[21, 50]$ | $[50, 100]$ | $[50, 100]$ | $[50, 100]$ | $[50, 100]$ |
| Batch Size | 256 | 8 | 10 | 6 | 10 | 10 |
| Epoch | 60 | 50 | 100 | 100 | 100 | 100 |

## H.3 EXPERIMENTS ON THE SECOND-ORDER TRANSFORMER

Table 11: Comparison of Pow-EMLR, ScalePow-EMLR and Pow-TMLR under the SoT-7 backbone on the ImageNet-1k dataset.

| Classifier | Top-1 Acc (%) | Top-5 Acc (%) |
|---|---|---|
| Pow-TMLR | 75.79 | 92.91 |
| ScalePow-EMLR | **76.14** | **93.18** |
| Pow-EMLR | 76.11 | 93.05 |

To further validate our findings, we follow Song et al. (2022b) to conduct experiments using the Second-order Transformer (SoT) (Xie et al., 2021) on the ImageNet-1k dataset. Specifically, we use the 7-layer SoT (SoT-7) architecture as the backbone network and train the model up to 250 epochs with a batch size of 512, keeping the other settings the same as Song et al. (2022b).

As shown in Tab. 11, Pow-EMLR still achieves similar performance to ScalePow-EMLR, but outperforms Pow-TMLR. These results further support our claim that tangent classifiers cannot adequately explain the matrix functions used in GCP, while the underlying mechanism can be better explained by our Riemannian perspective.

### H.4 Ablations on the bias reformulation

Recalling the vanilla Pow-EMLR $\langle A_k, S^\theta \rangle - b_k$, it can be rewritten as $\langle A_k, S^\theta - P_k \rangle$ according to Eq. (10), where $\langle P_k, A_k \rangle = b_k$. This reformulation is the key step to extending Euclidean MLR. Although this reformulation has shown success in different Riemannian MLRs (Ganea et al., 2018; Shimizu et al., 2020; Chen et al., 2024a;d; Nguyen & Yang, 2023), we conduct ablations on this reformulation. For simplicity, we set $P_k$ identical for all $k$; otherwise, it will bring a $[k, n, n]$ intermediate tensor for each $[n, n]$ covariance. We compare the following to MLR:

$$\text{Pow-EMLR: } \langle A_k, S^\theta \rangle - b_k, \tag{41}$$

$$\text{Pow-EMLR': } \langle A_k, S^\theta - P \rangle, \tag{42}$$

where $A_k, P \in \mathcal{S}^n$. Tab. 12 shows the results on all three fine-grained datasets. We observe that Pow-EMLR performs similarly to Pow-EMLR'.

Table 12: Comparison of Pow-TMLR, Pow-EMLR, and Pow-EMLR' on all three fine-grained datasets.

| Classifier | Air | | Birds | | Car | |
|---|---|---|---|---|---|---|
| | Top-1 Acc (%) | Top-5 Acc (%) | Top-1 Acc (%) | Top-5 Acc (%) | Top-1 Acc (%) | Top-5 Acc (%) |
| Pow-TMLR | 69.58 | 88.68 | 52.97 | 77.8 | 51.14 | 74.29 |
| Pow-EMLR' | 73.03 | 90.4 | 63.96 | 85.02 | 80.06 | 94.02 |
| Pow-EMLR | 72.07 | 89.83 | 63.29 | 84.66 | 80.43 | 94.15 |

## I Proof of Thm. 2

This proposition is mainly inspired by Chen et al. (2024a, Thm. 5). However, all the results by Chen et al. (2024a) require the metric to be a pullback metric from a *standard Euclidean space*, while the metric in our Thm. 2 is a pullback metric from *the SPD manifold*. Nevertheless, we still can reach similar theoretical results. We first recap RSGD and then begin to present our proof.

RSGD (Bonnabel, 2013) is formulated as

$$\bar{W} = \text{Exp}_W(-\gamma \Pi_W(\nabla_W f)) \tag{43}$$

where $\text{Exp}_W$ is the Riemannian exponential map at $W$, and $\Pi_W$ maps the Euclidean gradient $\nabla_W f$ to the Riemannian gradient, and $\gamma$ denotes learning rate.

We denote $(1, 0)$-EM as EM, and the metric tensor of it as $g^E$. Instead of providing an ad hoc proof exclusively for PEM, we present the following two more general lemmas.

**Lemma 7.** *Given a diffeomorphism $\phi : \mathcal{S}^n_{++} \to \mathcal{S}^n_{++}$, $\phi$ induces a pullback metrics on $\mathcal{S}^n_{++}$ from $\{\mathcal{S}^n_{++}, g^E\}$, denoted as $g^{\phi\text{-}E}$. The $g^{\phi\text{-}E}$-induced SPD MLR is*

$$p(y = k|S) \propto \exp\left[\langle \phi(S) - \phi(P_k), \phi_{*,I}(A_k) \rangle\right], \tag{44}$$

*where $S \in \mathcal{S}^n_{++}$ is an input feature, $P_k \in \mathcal{S}^n_{++}$ and $A_k \in \mathcal{S}^n$ are parameters for each class $k$.*

*Proof.* According to Chen et al. (2024d, Thm. 3.3), the Riemannian MLR based on $g^{\phi\text{-}E}$ is given as

$$p(y = k|S) \propto \exp\left[g^{\phi\text{-}E}_{P_k}(\text{Log}_{P_k} S, \text{PT}_{I \to P_k} A_k)\right]$$
$$= \exp\left[\langle \phi(S) - \phi(P_k), \phi_{*,I}(A_k) \rangle\right], \tag{45}$$

where Eq. (45) can be obtained by the properties of deformed metrics (Thanwerdas & Pennec, 2022, Tab. 2) and EM (Thanwerdas & Pennec, 2023, Tab. 3). □

Following the notations in Lem. 7, we have the following lemma.

**Lemma 8.** *Supposing $\phi_{*,I}$ is the identity map and each SPD parameter $P_k$ (Euclidean parameter $A_k$) in Eq. (44) is optimized by $g^{\phi\text{-}E}$-based RSGD (Euclidean SGD), the $g^{\phi\text{-}E}$-based SPD MLR is equivalent to a Euclidean MLR illustrated in Eq. (10) in the co-domain of $\phi$.*

*Proof.* We first show the projection operator $\Pi_P$ at $P \in \mathcal{S}_{++}^n$ under $g^{\phi\text{-E}}$, and then move on to the equivalence.

For any smooth function $f : \mathcal{S}_{++}^n \to \mathbb{R}$ on $\mathcal{S}_{++}^n$ endowed with $g^{\phi\text{-E}}$, we denote its Euclidean and Riemannian gradient at $P \in \mathcal{S}_{++}^n$ as $\nabla_P f$ and $\tilde{\nabla}_P f$, respectively. Then, for any $V \in T_P \mathcal{S}_{++}^n$, we have

$$\langle \tilde{\nabla}_P f, V \rangle_P = V(f) \Rightarrow \langle \phi_{*,P} \tilde{\nabla}_P f, \phi_{*,P} V \rangle = \langle \nabla_P f, V \rangle$$
$$\Rightarrow \Pi_P(\nabla_P f) = \phi_{*,P}^{-1} \circ (\mathrm{ad}(\phi_{*,P}))^{-1} (\nabla_P f), \tag{46}$$

where $\mathrm{ad}(\cdot)$ is the adjoint operator of the linear map.

According to Eq. (10), we define a Euclidean MLR in the codomain of $\phi$ as

$$p(y = k \mid S) \propto \exp(\langle \phi(S) - \bar{P}_k, \bar{A}_k \rangle), \text{ with } \bar{P}_k, \bar{A}_k \in \mathcal{S}^n. \tag{47}$$

We call this classifier $\phi$-EMLR.

Following Lem. 7, the SPD MLR under $g^{\phi\text{-E}}$ is

$$p(y = k \mid S) \propto \exp(\langle \phi(S) - \phi(P_k), \tilde{A}_k \rangle), \text{ with } P_k \in \mathcal{S}_{++}^n, \tilde{A}_k \in \mathcal{S}^n. \tag{48}$$

Supposing the SPD MLR and $\phi$-EMLR satisfying $\bar{P}_k = \phi(P_k)$. Other settings of the network are all the same, indicating the Euclidean gradients satisfying

$$\frac{\partial L}{\partial \bar{P}_k} = \frac{\partial L}{\partial \phi(P_k)}. \tag{49}$$

The updates of $\bar{P}_k$ in the $\phi$-EMLR is

$$\bar{P}_k' = \bar{P}_k - \gamma \frac{\partial L}{\partial \bar{P}_k}. \tag{50}$$

The updates of $P_k$ in the SPD MLR is

$$P_k' = \mathrm{Exp}_{P_k}(-\gamma \Pi_{P_k}(\nabla_{P_k} f))$$
$$= \phi^{-1} \left[ \phi(P_k) - \gamma (\mathrm{ad}(\phi_{*,P}))^{-1} \left( \frac{\partial L}{\partial P_k} \right) \right]. \tag{51}$$

Therefore $\phi(P_k')$ satisfies

$$\phi(P_k') = \phi(P_k) - \gamma (\mathrm{ad}(\phi_{*,P}))^{-1} \left( \frac{\partial L}{\partial P_k} \right)$$
$$= \phi(P_k) - \gamma (\mathrm{ad}(\phi_{*,P}))^{-1} \circ \mathrm{ad}(\phi_{*,P_k}) \left( \frac{\partial L}{\partial \phi(P_k)} \right)$$
$$= \phi(P_k) - \gamma \frac{\partial L}{\partial \phi(P_k)}$$
$$= \bar{P}_k'. \tag{52}$$

The second equation comes from the Euclidean chain rule of differential. Let $Y = \phi(X)$, then we have

$$\frac{\partial L}{\partial Y} : \mathrm{d}Y = \frac{\partial L}{\partial Y} : \phi_{*,X}(\mathrm{d}X)$$
$$= \mathrm{ad}(\phi_{*,X}) \left( \frac{\partial L}{\partial Y} \right) : \mathrm{d}X, \tag{53}$$

where $\cdot : \cdot$ means Frobenius inner product.

The equivalence of $\bar{A}_k$ and $\tilde{A}_k$ is obvious. Since both forward and backward processes of Eq. (47) and Eq. (48) are identical, by natural induction, the lemma can be proven. $\qquad\square$

When $\theta > 0$, simple computation shows that $(\theta, 1, 0)$-EM is the pullback metric of EM by $\phi_\theta$ with $\phi_{\theta*,I}$ as an identity map. According to Lems. 7 and 8, one can readily prove Thm. 2.

## J  ADDITIONAL DISCUSSIONS ON THM. 2

In Lem. 8, $\phi_{*,I}$ is required to be identity map. However, Lem. 8 can be extended into the case where $\phi_{*,I}$ is not the identity map, which will further extend Thm. 2 into the case of $\theta < 0$.

Following the notations in Lem. 7, let $\phi : \mathcal{S}_{++}^n \to \mathcal{S}_{++}^n$ be a diffeomorphism, whose differential at $I$, *i.e.*,$\phi_{*,I}$ is not an identity map. As $\phi$ is a diffeomorphism, the differential map $\phi_{*,I} : T_I \mathcal{S}_{++}^n \to T_{\phi(I)} \mathcal{S}_{++}^n$ is a linear isomorphism, *i.e.,*a bijection preserving linear operations. Therefore, we can identify $\tilde{A}_k = \phi_{*,I}(A_k)$ with $A_k$ in Eq. (44), and the SPD MLR under $g^{\phi\text{-E}}$ is simplified as

$$p(y = k|S) \propto \exp\left[\langle \phi(S) - \phi(P_k), \tilde{A}_k \rangle\right], \tag{54}$$

where $\tilde{A}_k \in \mathcal{S}^n$. As a direct corollary, all the proof in Lem. 8 can be transferred to the case where $\phi_{*,I}$ is not an identity.

**Corollary 9.** *Supposing $\phi : \mathcal{S}_{++}^n \to \mathcal{S}_{++}^n$ is a diffeomorphism and each SPD parameter $P_k$ (Euclidean parameter $A_k$) in Eq. (54) is optimized by $g^{\phi\text{-E}}$-based RSGD (Euclidean SGD), the $g^{\phi\text{-E}}$-based SPD MLR is equivalent to a Euclidean MLR in the co-domain of $\phi$.*

As a direct application of Cor. 9, Thm. 2 can be generalized into the case of $\theta < 0$. We generalize the definition of $\phi_\theta$ as

$$\phi_\theta(S) = \frac{1}{|\theta|} S^\theta, \forall S \in \mathcal{S}_{++}^n, \text{ with } \theta \neq 0. \tag{55}$$

Obviously, $\phi_\theta : \mathcal{S}_{++}^n \to \mathcal{S}_{++}^n$ is still a diffeomorphism. By Cor. 9, we can readily obtain the following results.

**Corollary 10.** *Under PEM with $\theta \neq 0$, optimizing each SPD parameter $P_k$ in Eq. (54) by PEM-based RSGD and Euclidean parameter $A_k$ by Euclidean SGD, the PEM-based SPD MLR is equivalent to a Euclidean MLR illustrated in Eq. (10) in the co-domain of $\phi_\theta(\cdot)$.*

Rahman et al. (2023) adopted the inverse of the covariance matrix for GCP. Cor. 10 indicates that our framework can also explain the underlying mechanism of the inverse by Rahman et al. (2023).

