# OpenReview forum: "Understanding Matrix Function Normalizations in Covariance Pooling through the Lens of Riemannian Geometry"
_ICLR.cc/2025/Conference — ICLR 2025 Poster_

### Official Review · Reviewer_2skh · 2024-10-31

**Soundness:** 2
**Presentation:** 3
**Contribution:** 2
**Rating:** 6
**Confidence:** 5

**Summary:**

This paper begins by exploring the motivation behind MPN-COV, focusing on how matrix power normalization (MPN) can be regarded as an approximation of geodesic distance (Log-E metric) for SPD matrices. Experimental results show there exist a significant discrepancy between MPN and Log-E metric.

Then, the authors try to study whether the theoretical framework used for matrix logarithmic normalization can explain matrix power normalization. They find that matrix power normalization maps SPD matrices to the tangent space (Euclidean space) at the identity matrix, and construct Pow-TMLR. But experiments reveals that Pow-TMLR shows spoor performance.

Finally, following by the works [Nguyen & Yang, 2023; Chen et al., 2024a;c] those expanded Euclidean MLR into SPD manifolds, this paper extends matrix logarithmic and matrix power normalization to SPD multinomial logistics regression. They construct ScalePow-EMLR based on this theory and demonstrate through experiments that this theoretical framework provides a coherent and unified explanation for the direct addition of Euclidean space classifiers after matrix power and logarithmic normalization.

**Strengths:**

The paper provides a different explanation on why matrix power normalization and matrix logarithmic normalization can be followed by a Euclidean space classifier.

This paper is well written.

**Weaknesses:**

1.	This paper provides an explanation based on SPD multinomial logistics regression for why matrix logarithmic normalization and matrix power normalization can be followed by Euclidean space classifiers. However, it seems not very clear why matrix power normalization significantly outperforms matrix logarithmic normalization under this framework. In other words, what are the significant performance differences between the two methods under the explanation framework of SPD MLR?

2.	As shown in Eqn. (15) and Eqn. (16), the authors extend matrix power normalization and matrix logarithmic normalization to SPD MLR. However, it seems not very clear why matrix power normalization and matrix logarithmic normalization can be regarded as Riemannian classifiers based on Eqn. (15) and Eqn. (16).

3.	As shown in Table 4, there only exists a difference of constant term between the formula of ScalePow-EMLR and one of Pow-TMLR. But, as compared in Tables 3, 5, and 6, there is a significant performance gap between ScalePow-EMLR and Pow-TMLR, where ScalePow-EMLR and Pow-EMLR are equivalent. For example, on the Cars dataset in Table 5, the Top-1 accuracy of ScalePow-EMLR is 80.31%, while that of Pow-TMLR is only 51.14%. I feel a bit confused why a constant term can bring significant performance gap? Could the authors give more explanation on this phenomenon?

**Questions:**

Please see paper weaknesses.

---

> ### Author Response · Authors · 2024-11-23
> **Response to Reviewer 2skh (Part 1)**
>
> We thank reviewer $\textcolor{red}{2skh}$ for the valuable feedback. Below is our detailed response. 😄
>
> ***
> ## **1. Analysis of the Empirical Difference between Matrix Power and Logarithm**
>
> - **Empirical Difference and Latent Geometries.** This paper demonstrates that matrix power and logarithm in GCP respect the Power-Euclidean Metric (PEM) and Log-Euclidean Metric (LEM), respectively. The empirical differences between them can thus be attributed to the underlying metrics. As shown in [a, Sec. 3.1], PEM converges to LEM as $\theta \to 0$, meaning PEM serves as a balanced metric between LEM and the Euclidean Metric ($\theta=1$), offering greater flexibility. GCP applications might benefit from this balancing effect. Supporting evidence from [b, Tabs. 4 and 7] also suggests that PEM may outperform LEM in certain tasks, highlighting its practical effectiveness.
> - **Treating Metrics as Hyperparameters.** Rather than focusing solely on matrix functions, our work uncovers the latent geometries they respect. A key insight from this work is that the choice of metric should be treated as a hyperparameter, as it is difficult to identify a universally optimal metric for all applications due to the non-convex nature of deep networks. The current empirical superiority of matrix power over logarithm may stem from PEM being better suited to the GCP application than LEM. Previous studies on the SPD manifold have shown that the optimal metric often varies across tasks [b-f]. We hope our work will inspire further exploration of alternative geometries on the SPD manifold to design more effective GCP classifiers.
> - **Numerical Issues of Matrix Logarithm.** Beyond theoretical considerations, the matrix logarithm can excessively stretch eigenvalues [g], which may lead to suboptimal performance in high-dimensional settings, such as those encountered in GCP. In contrast, the matrix power in PEM exhibits more moderate eigenvalue scaling. **Tab. B** illustrates a numerical comparison between matrix logarithm and square root in terms of eigenvalue behavior.
>
> | **Expression**              | $\frac{a}{b}$  | $\frac{\log(a)}{\log(b)}$ | $\frac{\sqrt{a}}{\sqrt{b}}$ |
> |-----------------------------|----------------|---------------------------|-----------------------------|
> | **Value**                   | $1,000,000$    | $-1$                      | $1,000$                    |

---

> ### Author Response · Authors · 2024-11-23
> **Response to Reviewer 2skh (Part 2)**
>
> ## **2. Equivalence of Eq. (15) or (16) to the SPD MLR**
>
> Let us first review the related equations. For each class $\forall k \in 1, \cdots, C$, we have:
> $$
> \textbf{Eq. (15): }  \left\langle \log(S) - \log(P _k), A _k \right\rangle \overset{\text{Prop. 5.1 [d]}}{\Longleftrightarrow}  \textbf{Log-EMLR: } \mathcal{F} \left( f _{\mathrm{vec}}  (\log(S)) \right),
> $$
> $$
> \textbf{Eq. (16): } \frac{1}{\theta}\left\langle S ^\theta-P _k ^\theta, A _k\right\rangle \overset{\text{Thm. 2}}{\Longleftrightarrow} \textbf{ScalePow-EMLR: } \mathcal{F}\left(f _{\text {vec }}\left(\frac{1}{\theta} S ^\theta\right)\right),
> $$
> $$
> \text{ScalePow-EMLR} \underset{\text{Empirically: Sec. 6}}{\overset{\text{Theoretically: App. G.1}}{\Longleftrightarrow}}\textbf{Pow-EMLR (Used in GCP): } \mathcal{F} \left(f _{\mathrm{vec}} \left(S ^\theta \right) \right),
> $$
> where $S$ is the input SPD matrix, $P _k \in \mathcal{S} _{++} ^n$ is an SPD parameter, and $A _k \in \mathcal{S} ^n$ is a symmetric matrix parameter. Here, $\mathcal{F}$ represents the FC layer, and $f _{\text{vec}}$ denotes the vectorization. Without loss of generality, we omit the softmax operation.
>
> The equivalence between Eq. (15) and Log-EMLR has been rigorously proven by [d, Prop. 5.1]. Our focus now shifts to Pow-EMLR. Specifically, we first prove the equivalence between Eq. (16) and ScalePow-EMLR (as presented in Thm. 2) and then demonstrate the equivalence between ScalePow-EMLR and Pow-EMLR both theoretically and empirically.
> - **Eq. (16) and ScalePow-EMLR.** As each $P_k$ in Eq. (16) is an SPD parameter, it is non-trivial to directly identify Eq. (16) with ScalePow-EMLR. Different from Euclidean parameters, $P_k$ requires Riemannian optimization, such as the Riemannian SGD or Riemannian gradient. However, Thm. 2 shows that when $P_k$ is optimized bt PEM-based Riemannian optimization, Eq. (16) and ScalePow-EMLR become equivalent. As proven in App. I, Eq. (16) is identical to ScalePow-EMLR during each optimization step.
> - **ScalePow-EMLR and Pow-EMLR.** The difference between ScalePow-EMLR and Pow-EMLR lies only in a scaling factor, which should have minor practical effects. Although the scaling factor affects the gradient, these two formulations can become equivalent under carefully tuned initialization and learning rates. For rigorous theoretical analysis, please refer to App. G.1.
> - **Conclusion.** We claim that Pow-EMLR is implicitly the SPD MLR described in Eq. (16), as it is theoretically equivalent to ScalePow-EMLR under scaled settings. Experimental results in Sec. 6 further validate this claim, demonstrating that Pow-EMLR achieves performance comparable to ScalePow-EMLR. We summarize this reasoning as follows:
> $$
> \text{Eq. (16)} \overset{\text{Thm. 2}}{\Longleftrightarrow} \text{ScalePow-EMLR} \underset{\text{Empirically: Sec. 6}}{\overset{\text{Theoretically: App. G.1}}{\Longleftrightarrow}} \text{Pow-EMLR (Used in GCP)}.
> $$

---

> ### Author Response · Authors · 2024-11-23
> **Response to Reviewer 2skh (Part 3)**
>
> ### **3. Explanations: Pow-EMLR vs. Pow-TMLR**
>
> $$
> \textbf{Tangent Classifier Pow-TMLR: } \mathcal{F}\left(f_{\text {vec }}\left(\frac{1}{\theta}\left(S^{\theta}-I\right)\right)\right),
> $$
> $$
> \textbf{Riemannian Classifier Pow-EMLR: } \mathcal{F}\left(f_{\text {vec }}\left(S^{\theta}\right)\right).
> $$
> As recapped above, Pow-EMLR and Pow-TMLR differ only by a constant transformation $f(X)=\frac{1}{\theta}\left(X-I\right)$. However, this difference introduces both theoretical and numerical disparities.
> - **Theoretical Explanations.** Pow-TMLR operates as a tangent classifier on the SPD manifold. By identifying the entire manifold with a single flat tangent space, it fails to respect the complex SPD geometry. In contrast, Pow-EMLR is equivalent to the SPD MLR, which is grounded in the Riemannian geometry of the SPD manifold. As such, Pow-EMLR naturally has theoretical advantages over the approximated tangent-based Pow-TMLR. Several studies have demonstrated the effectiveness of intrinsic Riemannian MLRs across various geometries, including SPD manifolds [b-d] and hyperbolic manifolds [h-j]. These findings serve as the main supporting evidence for the superiority of Pow-EMLR over Pow-TMLR.
> - **Numerical Explanations.** Beyond the explanation from Riemannian geometry, as discussed in App. G.2, although Pow-EMLR and Pow-TMLR can be aligned during the forward process (by adjusting initialization), their gradient backpropagation differs. This subtle difference makes Pow-EMLR and Pow-TMLR distinct in the training dynamics of the network, further explaining the observed performance gap. Please refer to App. G.2 for more details.
>
> ***Remark:** Initially, during this work, we attempted to use tangent classifiers to explain the role of matrix functions in GCP. Despite fine-tuning efforts, Pow-TMLR usually failed to achieve performance comparable to Pow-EMLR. This counterintuitive result also puzzled us. However, we believe there must be some other more fundamental reasons. When we delved into this myth deeper, we finally discovered that Pow-EMLR is implicitly a Riemannian MLR, which is a fundamentally different mechanism from the tangent perspective. This finding explains not only the mechanism underlying matrix functions in GCP but also the unexpected performance gap observed. We hope our findings will inspire further research of GCP  from a geometric perspective.*
>
> ***
> ## References
> > [a] The geometry of mixed-Euclidean metrics on symmetric positive definite matrices
> >
> > [b] RMLR: Extending multinomial logistic regression into general geometries
> >
> > [c] Building neural networks on matrix manifolds: A Gyrovector space approach
> >
> > [d] Riemannian multinomial logistics regression for SPD neural networks
> >
> > [e] A Lie group approach to Riemannian batch normalization
> >
> > [f] The Gyro-structure of some matrix manifolds
> >
> > [g] Why approximate matrix square root outperforms accurateSVD in global covariance pooling
> >
> > [h] Hyperbolic neural networks
> >
> > [i] Hyperbolic neural networks++
> >
> > [j] Fully hyperbolic convolutional neural networks

---

> > ### Comment · Reviewer_2skh · 2024-11-26
> > **Thanks for your response**
> >
> > Thanks for your response, which addresses my concern on Eqn. (15) and Eqn. (16). But I still feel a bit confused about the significant performance differences between matrix logarithmic normalization and matrix power normalization under the explanation framework of SPD MLR. The empirical difference and numerical issues of matrix logarithm have been studied in previous works, so I sincerely would like to know what new insights about explanation are on why matrix power normalization is better than matrix logarithmic normalization under the framework of SPD MLR.
> >
> > Besides, App. G.2 only shows the difference between Pow-TMLR and Pow-EMLR, which does not clearly account for why Pow-TMLR is significantly inferior to Pow-EMLR with introducing a constant term. It is encouraged to be discussed in detail in the revision.

---

> ### Author Response · Authors · 2024-11-26
> **Additional Response**
>
> ***
> Thanks for the instant reply and further comments! We make the following explanations.
>
> ***
> ## **1. Novel Insights: From Matrix Functions to Matrix Geometries**
>
> We discover that matrix log and pow in GCP respect Log Euclidean Metric (LEM) and Power Euclidean Metric (PEM), respectively. Therefore, the empirical difference can be attributed to the difference in their Riemannian metrics. Apart from preserving several properties of LEM, PEM offers greater flexibility.
>
> - **Shared Properties:** PEM shares several key properties with LEM. Both are $\mathrm{O}(n)$-invariant [A, Tab. 2], a crucial property for modeling covariance matrices. Additionally, their Riemannian computations (e.g., Riemannian logarithm, exponentiation, and parallel transport) are similar, modulo the computation of matrix log or power.
> - **Flexible Deformation:** PEM is a deformed metric of LEM. It approaches LEM with power factor $\theta \rightarrow 0$.
>
> The above suggests that PEM functions as a balanced metric, maintaining the essential properties of LEM while introducing greater flexibility. This added flexibility could become particularly advantageous in high-dimensional settings, such as those encountered in GCP applications, potentially explaining why PEM outperforms LEM in these cases.
>
> **Remark:** [$\mathrm{O}(n)$-invariance] When a random vector $x$ with covariance matrix $\Sigma$ is transformed by $O x$, the resulting covariance matrix becomes $O \Sigma O^\top$. $\mathrm{O}(n)$-invariance is to characterize this relation between covariance matrices:
> \begin{equation}
> d(P,Q) = d(O P O^\top,O Q O^\top), \quad \forall P, Q \in \mathcal{S} _{++} ^n, \forall O \in \mathrm{O}(n),
> \end{equation}
> where $d$ denotes the geodesic distance.
>
> ***
> ## **2. Pow-TMLR vs. Pow-EMLR --> Tangent Classifiers vs. Riemannian Classifiers**
>
> Although the numerical expressions of Pow-TMLR and Pow-EMLR differ by a constant transformation, they differ fundamentally in theory: Pow-TMLR is a tangent classifier, whereas Pow-EMLR is a Riemannian classifier.
>
>
> - **Tangent Classifier:** The tangent classifier treats the entire manifold as a single tangent space at the identity matrix. When mapping data into this tangent space, critical structural information, such as distances and angles, cannot be preserved. This distortion undermines classification performance. In contrast, Riemannian MLR is constructed based on Riemannian geometry, fully respecting the manifold's geometric structure.
>
> - **Tangent as a Special Case of Riemannian Classifier:** The tangent classifier can be seen as a reduced case of the Riemannian classifier. For example, consider Eq. (16) (PEM):
> \begin{equation}
> \textbf{Eq. (16): } P(y=k|S) \propto \exp \left[\frac{1}{\theta}\left\langle S ^\theta-P _k ^\theta, A _k \right\rangle\right], \quad \forall k =1, \cdots, C.
> \end{equation}
> When all SPD parameters $P_k$ are set to the fixed identity matrix, Eq. (16) exactly corresponds to Pow-TMLR.
>
> In summary, the Riemannian classifier enjoys significant theoretical advantages over the tangent classifier while incorporating the tangent classifier as a special case.
>
> **Remark:** We have added the above discussion into App. G.3 in our revised manuscript.
>
> ***
> ## **Further Comments Are Very Welcome**
>
> If you have any additional comments, we are happy to discuss them further.
>
>
> ***
> ## **References**
> > [A] RMLR: Extending multinomial logistic regression into general geometries

---

> > ### Comment · Reviewer_2skh · 2024-11-27
> > **Thanks for your reply**
> >
> > Thanks for your reply, and I am grateful to that the authors are willing to include more details on the difference between Pow-TMLR and Pow-EMLR.
> >
> > Besides, in my opinion, discussion on Log Euclidean Metric (LEM) and Power Euclidean Metric (PEM) from the perspective of Riemannian classifier is core contribution of this work, and I believe it may bring some different insights into the observation on superiority of PEM over LEM in the context of deep architectures. But the statement that PEM is a more flexible metric than LEM seems not new, which also seems not an exclusive finding under framework of Riemannian classifier. Meanwhile, many previous works show PEM with power of 0.5 generally achieves best performance across various tasks and deep architectures. Therefore, I am eagerly looking forward to some new findings on the differences between Log Euclidean Metric (LEM) and Power Euclidean Metric (PEM) from the perspective of Riemannian classifier.
> >
> > Additionally, if the authors could show what favorable properties of a good metric are under framework of Riemannian classifier, it may attract much more attention from the readers.
> >
> > The above questions may not be easily addressed in current revision, which are very welcome to be discussed in future work.

---

> ### Author Response · Authors · 2024-11-27
> **Thanks for the positive feedback!**
>
> Thanks for your further reply and insightful comments!
>
> We agree that our core contribution lies in uncovering the underlying mechanism of the Riemannian classifier. We will emphasize this point in the revised introduction and point to the detailed discussion in Sec. 5.3. The choice of a power of 0.5 is indeed more favorable for GCP applications, especially in the large-scale setting. While the reason why power=0.5 performs best is not fully addressed, our work sheds light on the mechanism of the Riemannian classifier, providing a novel perspective. In future work, we plan to delve deeper into understanding why PEM, particularly with power=0.5, outperforms LEM.
>
> We would like to discuss more about the desired properties of Riemannian metrics and future directions of manifold learning beyond SPD manifolds:
>
> ### **Geometries within the SPD Manifold**
> Three properties of metrics on the SPD manifold are particularly crucial: invariance, closed-form Riemannian operators, and isometry to the Euclidean metric. These properties enhance the representation power, ease of implementation, and computational efficiency of Riemannian classifiers.
> - **Invariance**, such as $\mathrm{O}(n)$-invariance, is a critical property for modeling covariance matrices. Metrics with $\mathrm{O}(n)$-invariance enable the construction of more powerful Riemannian classifiers. In addition to LEM and PEM, the recently developed Bures-Wasserstein metric [A, Tab. 6] also possesses $\mathrm{O}(n)$-invariance. Moreover, as shown in [A, Tab. 5], the classic Affine Invariant metric has been generalized into a two-parameter $\mathrm{O}(n)$-invariant metric. Riemannian classifiers under these metrics have closed-form expressions [B, Thm. 4.2]. Exploring the adaptation of such classifiers to GCP applications could be an interesting direction.
> - **Closed-form expressions** enable the practical development and implementation of Riemannian classifiers. All seven metrics discussed in Sec. 2 possess closed-form expressions for their Riemannian operators.
> - **Isometry to the Euclidean metric** facilitates fast and simple computations. Riemannian computations under such metrics reduce to Euclidean computations, modulo the calculation of isometric diffeomorphisms [C, Tab. 2]. This property could significantly simplify the final expression of Riemannian classifiers, making them well-suited for GCP applications.
>
> ### **Geometries beyond the SPD Manifold**
>
> Beyond SPD matrices such as covariance, correlation matrices can be viewed as normalized SPD matrices, serving as natural, compact alternatives to SPD matrices in statistical analysis. Recently, these structured matrices have been recognized as a manifold, and several Riemannian metrics have been identified [D, E]. These metrics enjoy the three properties mentioned above. Therefore, it is promising to develop Riemannian classifiers based on this manifold for GCP applications.
>
> Thanks again for the encouraging feedback. We will explore both geometries of and beyond SPD manifolds and find out more in our future work!
>
> ***
> ## **References**
> > [A] O(n)-invariant Riemannian metrics on SPD matrices
> >
> > [B] RMLR: Extending Multinomial Logistic Regression into General Geometries
> >
> > [C] The geometry of mixed-Euclidean metrics on symmetric positive definite matrices
> >
> > [D] Theoretically and computationally convenient geometries on full-rank correlation matrices 2022
> >
> > [E] Permutation-invariant log-Euclidean geometries on full-rank correlation matrices 2024

---

### Official Review · Reviewer_24iT · 2024-10-31

**Soundness:** 3
**Presentation:** 3
**Contribution:** 2
**Rating:** 6
**Confidence:** 3

**Summary:**

As global covariance pooling (GCP) becomes an increasingly important component in modern deep neural networks, a discrepancy remains between theoretical principles and practical applications of the normalization techniques used in GCP, especially regarding the matrix logarithm and matrix power. This work aims to bridge this gap by offering a theoretical understanding from a Riemannian geometry perspective. Through a combination of theoretical analysis and empirical evaluations, the study shows that the working mechanism of these matrix functions can be attributed to the Riemannian classifiers they implicitly respect.

**Strengths:**

1. This work identifies the discrepancy between the theory and practice of matrix normalization in GCP and addresses it by providing comprehensive theoretical explanations and underlying rationale for using Euclidean classifiers.
2. The study is technically sound, combining theoretical proof with empirical evidence.
3. Experiments on several real-world datasets are conducted to support the theoretical claims.
4. Overall, this paper is well-written and thoughtfully structured.

**Weaknesses:**

1. I am concerned that the theoretical contribution of this work appears to have a somewhat narrow application scope in its current presentation, as it focuses specifically on matrix normalization methods in GCP for deep neural networks, particularly on the matrix logarithm and matrix power. I suggest that the authors further clarify the practical significance of matrix normalization and the presented theories.
2. There are some other matrix normalization methods in GCP beyond matrix logarithm and matrix power. Popular choices [1] include matrix logarithm, element-wise power, matrix square-root and matrix power normalization. Is the element-wise power normalization applicable to the presented theories? Does it require additional analysis?
3. In Figure 1, in addition to showing the gap between LEM and PEM, it would be beneficial to highlight the consequences of this gap.


[1] Wang, Qilong, et al. "What deep CNNs benefit from global covariance pooling: An optimization perspective." Proceedings of the IEEE/CVF Conference on Computer Vision and Pattern Recognition. 2020.

**Questions:**

1. I am concerned that the theoretical contribution of this work appears to have a somewhat narrow application scope in its current presentation, as it focuses specifically on matrix normalization methods in GCP for deep neural networks, particularly on the matrix logarithm and matrix power. I suggest that the authors further clarify the practical significance of matrix normalization and the presented theories.
2. There are some other matrix normalization methods in GCP beyond matrix logarithm and matrix power. Popular choices [1] include matrix logarithm, element-wise power, matrix square-root and matrix power normalization. Is the element-wise power normalization applicable to the presented theories? Does it require additional analysis?
3. In Figure 1, in addition to showing the gap between LEM and PEM, it would be beneficial to highlight the consequences of this gap.


[1] Wang, Qilong, et al. "What deep CNNs benefit from global covariance pooling: An optimization perspective." Proceedings of the IEEE/CVF Conference on Computer Vision and Pattern Recognition. 2020.

---

> ### Author Response · Authors · 2024-11-23
> **Response to Reviewer 24iT**
>
> We thank Reviewer $\textcolor{blue}{24iT}$ for valuable comments. Below, we address the comments in detail. 😄
>
> ***
> ## **1. Practical Significance: From Matrix Functions to Various Latent Geometries**
>
> - **From Matrix Functions to Latent Geometries.** Previous work primarily views the matrix functions in GCP as numerical normalization methods. While this numerical interpretation provides empirical insights, it lacks an in-depth theoretical foundation. Covariance matrices, however, lie on the SPD manifold, which is endowed with rich Riemannian structures. A natural idea is that Riemannian features may benefit more from operations grounded in Riemannian geometry rather than Euclidean manipulations. This idea aligns with findings from recent studies across various geometries [a-i]. Inspired by these advancements, we collectively interpret matrix power and logarithm as special SPD MLRs under the Power-Euclidean Metric (PEM) and Log-Euclidean Metric (LEM), respectively. In this way, the empirical performance of power and log can be attributed to the metrics they respect. We expect this novel view can shed light on the GCP research.
> - **From PEM and LEM to Other Geometries.** Our discovery hints at the potential to design classifiers in GCP by leveraging Riemannian classifiers. As summarized in Sec. 2, there are five other metrics on the SPD manifold beyond PEM and LEM. We expect that our work could inspire the development of GCP based on these alternative metrics.
>
> The above impact has been briefly mentioned in our conclusion. To further highlight our significance, we have added the following to the introduction (L 93-95):
> - *We expect our work to pave the way for a deeper theoretical understanding of GCP from a Riemannian perspective and inspire more research to explore the rich SPD geometries for more effective GCP applications.*
>
> ***
> ## **2. Our Work Also Explains the Empirical Superiority of Matrix Power over Element-Wise Power**
>
> As demonstrated in Thm. 2, matrix power, including the matrix square root, implicitly constructs the intrinsic SPD MLR under the Power-Euclidean Metric (PEM). In contrast, element-wise power applied to SPD matrices is merely a numerical trick. It neither connects to nor respects any of the seven Riemannian metrics (reviewed in Sec. 2) on the SPD manifold. Previous works [a-i] have demonstrated that Riemannian features benefit from Riemannian operations on different geometries. Therefore, this theoretical distinction provides a novel explanation for the empirical superiority of matrix power over element-wise power observed in previous work [j].
>
> ***
> ## **3. Changed Caption of Fig. 1**
>
> Fig. 1 is intended to refute the discussions in L 53-54. Previous works [j,k] claim that since the distances under PEM and LEM are similar, and the co-domain of the matrix logarithm is a Euclidean space, it is reasonable to use a Euclidean classifier after the matrix power. However, Fig. 1 demonstrates a significant gap between the distances under PEM ($\theta=0.5$) and LEM, challenging the validity of this assumption.
>
> Thank you for the suggestion. We have revised the caption of Fig. 1 to clarify the implications of this observed gap:
>
> - *This indicates that matrix power is not proximate to LEM for classification under the widely used $\theta=0.5$.*
>
> ***
> ## **References**
> > [a] RMLR: Extending multinomial logistic regression into general geometries
> >
> > [b] Building neural networks on matrix manifolds: A Gyrovector space approach
> >
> > [c] Riemannian multinomial logistics regression for SPD neural networks
> >
> > [d] A Lie group approach to Riemannian batch normalization
> >
> > [e] The Gyro-structure of some matrix manifolds
> >
> > [f] Hyperbolic neural networks
> >
> > [g] Hyperbolic neural networks++
> >
> > [h] Fully hyperbolic convolutional neural networks
> >
> > [i] Riemannian residual neural networks
> >
> > [j] What deep CNNs benefit from global covariance pooling: An optimization perspective
> >
> > [k] Is second-order information helpful for large-scale visual recognition?

---

> > ### Comment · Reviewer_24iT · 2024-11-27
> >
> > Dear authors,
> >
> > Thank you for your response. It addressed most of my concerns and I will consider raising my score.
> >
> > Best,
> > Reviewer

---

> > > ### Author Response · Authors · 2024-11-27
> > > **Thanks for increasing the score!**
> > >
> > > Thanks for the positive feedback and increasing the score! You insightful comments do help us improve the paper.

---

### Official Review · Reviewer_JvSJ · 2024-11-01

**Soundness:** 2
**Presentation:** 3
**Contribution:** 3
**Rating:** 6
**Confidence:** 4

**Summary:**

This paper analyzes classifiers for normalized covariance-pooled features. Specifically, the authors explore a rationale behind the effective linear classifier (Euclidean MLR) directly applied to square-rooted covariance features. By analyzing empirical performance results and theoretical formulations of Euclidean MLR and SPD MLR, this paper finds out that the Euclidean MLR of square-rooted covariance features is well explained by the SPD MLR on power-Euclidean metric. The experimental results on image classification using global covariance pooling (GCP) validates the claim on various settings.

**Strengths:**

+ The theoretical stuffs regarding SPD Riemannian geometry are insightful and the empirical performance comparison in Tab.3 is interesting.
+ Practical performance analysis is conducted on various types of datasets and networks.

**Weaknesses:**

- The discussion to induce SPD MLR in Sec.5 sounds a bit strange. In Sec.5, a Euclidean-MLR is reformulated as $\langle a,x\rangle - b \Rightarrow \langle a,x-p\rangle$ by re-parameterizing the bias as $b=\langle a,p\rangle$. If such re-parameterization is acceptable, one can more directly conclude that a Tangent-MLR is equivalent to the Euclidean-MLR by using the re-parameterization of TMLR: $\langle a,x-1\rangle -b \Rightarrow \langle a,x-p'\rangle$: EMLR, where $b=\langle a,p'-1\rangle$. It skips SPD-MLR and thus is a different conclusion from this paper. Therefore, the authors' claim seems to be less convincing.

- On the other hand, from a viewpoint of classifier formulation, Pow-EMLR is equivalent to Pow-TMLR since $\langle a,x\rangle -b=\langle a,x-1\rangle -b'$ with one-to-one correspondence between the parameters $(a,b)$ and $(a,b')$ via $b=b'+\langle  a,1\rangle$. Thus, the performance difference shown in Tab.3 comes from training dynamics only, especially in terms of the bias. This fundamental issue is not mentioned in this paper and is less connected to the classifier model (e.g., SDP-MLR) that this paper focuses on. As one could *guess* that disentanglement between classifier weights and a bias is preferable for training, it would be more interesting to delve into the training processes with much attention to the bias in the framework of covariance feature.

- This paper insists that an SDP-MLR is a fundamental classifier for covariance features. To validate the claim, it is necessary to compare the SDP-MLR with Tangent-MLR and practical EMLR for respective metrics of LEM and PEM (and optionally LCM). Related to the above-mentioned bias issue, the SDP-MLR would exhibit different training dynamics from the EMLR $\langle a,x\rangle -b$, resulting in different performance. Thus, Tab.5 should contain results of SDP-MLR directly applied to covariance features.

- The presentation of this paper is not so clear since it introduces less-relevant metrics in Sec.2, leading to messy notations (parameters) and rather complicated discussions. It would be enough to present three metrics of Log-Euclidean and Power-Euclidean; if this paper finds efficiency of Log-Cholesky metric for covariance features, it might be nice to contain LCM as an additional contribution.

**Questions:**

See the above-mentioned weak points.

---

> ### Author Response · Authors · 2024-11-23
> **Response to Reviewer JvSJ (Part 1)**
>
> We thank Reviewer $\textcolor{green}{JvSJ}$ for the careful review and the suggestive comments. Below, we address the comments in detail.😄
>
> ***
> ## **1. Reparameterization: manifold counterparts of $\langle a_k, x-0 \rangle - b_k$ and $\langle a_k, x-p_k \rangle$ are significantly different**
>
> Given $x \in \mathbb{R} ^n$, the two reparameterizations of $\langle a_k, x \rangle - b_k$ are equivalent in Euclidean space:
>
> $$
> \textbf{Eq. (A): } \langle a_k, x - 0 \rangle - b_k, \quad \textbf{Eq. (B): } \langle a_k, x - p_k \rangle,
> $$
> with $a _k \in \mathbb{R} ^n$, $p _k \in \mathbb{R} ^n$, and $b _k \in \mathbb{R}$.
>
> However, their extensions to manifolds are significantly different and lead to fundamentally distinct interpretations.
>
> ### **Reformulation via Eq. (A) is the tangent classifier discussed in Sec. 4**
> Given an SPD matrix $S \in \mathcal{S} _{++} ^n$, extending Eq. (A) to the SPD manifold $\mathcal{S} _{++} ^n$ is
>
> $$
> \langle a _k, x - 0 \rangle - b _k \Rightarrow \langle \mathrm{Log} _I(S), A _k \rangle_I - b _k, \quad \forall k = 1, \ldots, C, \quad \textbf{Eq. (C)}
> $$
> where $I$ is the identity matrix, $A _k \in T _I \mathcal{S} ^n _{++} \cong \mathcal{S} ^n$, and $b _k \in \mathbb{R}$. Here, $\mathcal{S} ^n$ is the Euclidean space of symmetric matrices, while $\langle , \rangle _{I}$ and $\mathrm{Log} _I$ are the Riemannian metric and logarithm at $I$. Eq. \(C\) corresponds to mapping $S \in \mathcal{S} ^n _{++}$ to the tangent space $T_I \mathcal{S} ^n _{++}$ using the Riemannian logarithm $\mathrm{Log} _I(S)$, followed by applying a Euclidean FC layer in the tangent space.
>
> Under PEM and LEM on the SPD manifold, Eq. \(C\) exactly reduces to Pow-TMLR and Log-EMLR:
>
> - **Pow-TMLR:** $\mathcal{F} \left(f _{\mathrm{vec}} \left( \frac{1}{\theta} \left(S ^\theta - I \right) \right) \right)$,
> - **Log-EMLR:** $\mathcal{F} \left(f _{\mathrm{vec}} \left( \log(S) \right) \right),$
> where $\mathcal{F}$ denotes the FC layer and $f _{\mathrm{vec}$ is the vectorization.
>
> Comparing this to Pow-EMLR in GCP:
>
> $$
> \textbf{Pow-EMLR: } \mathcal{F} \left(f_{\mathrm{vec}} \left(S^\theta \right) \right),
> $$
>
> Pow-TMLR and Pow-EMLR are quite similar. However, experimental results in Tabs. 3 and 5 consistently show that Pow-TMLR performs worse than Pow-EMLR. This observation suggests a more fundamental mechanism underlying the matrix functions in GCP. The above is what we discussed in Sec. 4.
>
> ### **Riemannian MLR via Eq. (B) is different from the tangent MLR via Eq. (A)**
>
> Extending Eq. (B) to the SPD manifold results in a fundamentally different formulation, leading to **SPD MLRs** [a, Thm. 3.3]:
> $$
> \langle a _k, x - p _k \rangle \Rightarrow \langle \mathrm{Log} _ {P _k}(S), \widetilde{A} _k \rangle _{P _k}, \quad \forall k = 1, \ldots, C, \quad \textbf{Eq. (D)}
> $$
> where $P _k \in \mathcal{S} ^n _{++}$ and $\widetilde{A} _k \in T _{P _k} \mathcal{S} ^n _{++}$ are the parameters. It can hardly be said that Eq. \(C\) and (D) are equivalent, although their Euclidean prototypes are equivalent. A simple intuition is that there are multiple tangent plain in Eq. (D), while Eq. \(C\) only has a single tangent space. After several steps of simplification, the SPD MLR under PEM and LEM [c, Thm. 4.2] for the SPD input $S \in \mathcal{S} ^n _{++}$ can be expressed as follows:
>
> - **LEM-based SPD MLR**:
>   $$
>   p(y = k \mid S) \propto \exp \left( \left\langle \log(S) - \log(P_k), A_k \right\rangle \right), \quad \quad \textbf{Eq. (E)}
>   $$
>   where $P_k \in \mathcal{S}^n_{++}$, and $A_k \in \mathcal{S} ^n$.
>
> - **PEM-based SPD MLR**:
>   $$
>   p(y = k \mid S) \propto \exp \left( \frac{1}{\theta} \left\langle S^\theta - P_k^\theta, A_k \right\rangle \right), \quad \quad \textbf{Eq. (F)}
>   $$
>   where $\theta > 0$, $P_k \in \mathcal{S}^n_{++}$, and $A_k \in \mathcal{S} ^n$.
>
> While SPD MLRs differ from tangent classifiers, they are also distinct from the Pow-EMLR and Log-EMLR used in GCP, as there is an SPD parameter $P _k$ for each class. However, our Thm. 2 demonstrates that the matrix power in GCP implicitly respects the SPD MLR under PEM. Similar results also hold for the matrix logarithm and LEM [b, Prop. 5.1]. This theoretical insight bridges the gap between practical GCP matrix functions and SPD MLRs.
>
> ## **Under trivial bias, Riemannian MLR becomes the tangent classifier**
>
> When the bias in Eq. (D) is set as the fixed "zero" element (identity matrix), the Riemannian MLR reduces to the tangent classifier described in Eq. (C). Unlike in Euclidean space, the bias plays a more critical role in Riemannian MLR, as it determines the anchor point of the tangent space corresponding to each class, which is sourced back to the definition of margin hyperplane [a, Eq. (5)] or [e, Eq. (22)].
>
> (To be continued)

---

> ### Author Response · Authors · 2024-11-23
> **Response to Reviewer JvSJ (Part 2)**
>
> ## **Following up on Response 1**
> ### **Further clarifications are more than welcome**
>
> As opposed to tangent classifiers (e.g., Eq. \(C\)) that approximate the manifold as a flat space, the SPD MLR Eq. (D) resorts to the idea of margin distance to the hyperplane characterizing each class [a, Eqs. (4-5)], which has shown success in different geometries, including SPD [a-c], Symmetric Positive Semi-Definite (SPSD) [d], and hyperbolic [e-f] manifolds.
>
> If you are interested in delving further into extending the Euclidean MLR Eq. (B) into manifold [a-g] or more explanations on Thm. 2, we would be more than happy to provide additional clarifications.
>
> ## **2. Bias is not disentangled.**
>
> In the original SPD MLRs Eqs. (E-F), $P _k$ is the biasing parameter. Thm. 3.2 demonstrates that Eq. (E) is equivalent to
> $$
> \textbf{ScalePow-EMLR: } \mathcal{F} \left(f _{\mathrm{vec}} \left( \frac{1}{\theta} S ^{\theta} \right) \right),
> $$
> where $\mathcal{F}$ is the standard FC layer. In our code, $\mathcal{F}$ in Pow-EMLR, Pow-TMLR, and ScalePow-EMLR are all implemented by `torch.nn.Linear`. There is no separate manipulation of the bias. The following is our pseudo-code.
>
> ```python
> # Pseudocode for Covariance Pooling and ResNet with Covariance Pooling
>
> import torch.nn as nn
>
> # Covariance Pooling Module
> class CovPooling(nn.Module):
>     def __init__(self, args):
>         """
>         Initializes the CovPooling module.
>         Args:
>             args: Configuration or arguments required for initialization.
>         """
>
>     def forward(self, x):
>         """
>         Forward pass for Covariance Pooling.
>         Args:
>             x: Input feature tensor.
>         Returns:
>             x_vec: Vectorized representation of the transformed SPD matrix.
>         """
>         # Step 1: Apply covariance pooling
>         x = self._cov_pool(x)
>
>         # Step 2: Compute matrix square root
>         x_sqrt = self._sqrtm(x)
>
>         # Step 3: Apply transformations
>         if self.transformed_mode == 'Pow-TMLR':
>             # $\frac{1}{\theta}=2$ for sqrt; Self.I is the identity matrix
>             tmp = 2 * (x_sqrt - self.I)
>             x_transformed = tmp
>         elif self.transformed_mode == 'ScalePow-EMLR':
>             # $\frac{1}{\theta}=2$ for sqrt
>             x_transformed = 2 * x_sqrt
>         elif self.transformed_mode == 'Pow-EMLR':
>             x_transformed = x_sqrt  # For Pow-EMLR
>
>         # Step 4: Vectorize the resulting matrix
>         x_vec = self.vec(x_transformed)
>
>         # Step 5: Return the vectorized matrix
>         return x_vec
>
> # ResNet with Covariance Pooling
> class ResNetCovPooling(nn.Module):
>     def __init__(self, args, fc_input_dim, num_classes):
>         """
>         Initializes the ResNetCovPooling model.
>         Args:
>             args: arguments for ResNet and CovPooling
>             fc_input_dim is the dim of vectorized SPD matrices.
>             num_classes: Number of output classes.
>         """
>         super(ResNetCovPooling, self).__init__()
>         # Step 1: Set up backbone, representation, and classifier
>         self.feature = ResNet(args.ResNet)  # ResNet backbone
>         self.representation = CovPooling(args.CovPooling)  # Covariance pooling representation
>         self.classifier = nn.Linear(fc_input_dim, num_classes)  # Fully connected layer
>
>     def forward(self, x):
>         """
>         Forward pass for the ResNetCovPooling model.
>         Args:
>             x: Input feature tensor.
>         Returns:
>             out: Class predictions.
>         """
>         # Step 1: Extract features using ResNet backbone
>         x = self.feature(x)
>
>         # Step 2: Apply Covariance Pooling for representation
>         x = self.representation(x)
>
>         # Step 3: Classify the representation
>         out = self.classifier(x)
>
>         # Step 4: Return class predictions
>         return out
> ```

---

> > ### Author Response · Authors · 2024-11-23
> > **Response to Reviewer JvSJ (Part 3)**
> >
> > ## **3. Tab. 5: ScalePow-EMLR respects the SPD MLR**
> >
> > As summarized by Tab. 4, ScalePow-EMLR respects the SPD MLR, while Pow-TMLR and Cho-TMLR correspond to the tangent classifier. As shown in Tab. 5, Pow-EMLR outperforms Pow-TMLR while achieving comparable performance to ScalePow-EMLR. Considering that ScalePow-EMLR differs from Pow-EMLR only by a scaling factor, this result supports our claim that the working mechanism of matrix functions in GCP aligns with the SPD MLR.
> >
> > ***
> > ## **4. Sec. 2 is preparing for a complete refutation to the tangent classifier perspective.**
> >
> > Log-EMLR is widely used as a classifier to map SPD matrices into the tangent space for classification [h-j]. Inspired by this, we naturally sought to find a similar mechanism for the matrix power. Therefore, Sec. 2 reviews seven different metrics on the SPD manifold, which represent all the metrics with closed-form Riemannian logarithms. Based on this foundation, Sec. 4 comprehensively presents all the tangent classifiers. Tab. 2 concludes that Pow-TMLR respects four different metrics and shares a similar formulation with Pow-EMLR. However, experimental results show that Pow-TMLR performs worse than Pow-EMLR. This discrepancy indicates that the tangent classifier perspective fails to explain the working mechanism of matrix functions in GCP.
> >
> > We believe Sec. 2 is essential to thoroughly refute the tangent classifier perspective. Due to the complexity of the manifold, considering only a subset of metrics would not provide a convincing refutation. However, we are open to compressing parts of Sec. 2 and relocating some of the extended discussion to the appendix to streamline the main text while maintaining the critical context.
> >
> > ***
> > ## References
> > > [a] RMLR: Extending Multinomial Logistic Regression into General Geometries
> > >
> > > [b] Riemannian Multinomial Logistics Regression for SPD Neural Networks
> > >
> > > [c] Building neural networks on matrix manifolds: A Gyrovector space approach
> > >
> > > [d] Matrix manifold neural networks++
> > >
> > > [e] Hyperbolic neural networks
> > >
> > > [f] Hyperbolic neural networks++
> > >
> > > [g] Fully hyperbolic convolutional neural networks for computer vision
> > >
> > > [h] A Riemannian network for SPD matrix learning
> > >
> > > [i] SPD domain-specific batch normalization to crack interpretable unsupervised domain adaptation in EEG
> > >
> > > [j] Geomnet: A neural network based on Riemannian geometries of SPD matrix space and Cholesky space for 3D skeleton-based interaction recognition

---

> > > ### Comment · Reviewer_JvSJ · 2024-11-27
> > >
> > > Thanks for your response.
> > > Though, you seem to repeat the same claim presented in the paper and to miss answering clearly to my concerns.
> > >
> > > ### 1.
> > > I can clarify my first concern as follows.
> > >
> > > 1. Pow-EMLR: $\langle A, S^\theta\rangle - b$
> > > 2. Pow-EMLR': $\langle A, S^\theta-P\rangle =\langle A, S^\theta\rangle - (\langle A,P\rangle)=\langle A, S^\theta\rangle - b'$
> > > 3. Pow-EMLR'': $\langle A, S^\theta-I\rangle - b = \langle A, S^\theta\rangle - (b+\langle A,I\rangle)=\langle A, S^\theta\rangle - b''$
> > > 4. Pow-TMLR: $\langle A, S^\theta-I\rangle - b$
> > >
> > > Following the discussion in the paper, I think all these formulations seem to be *equivalent*.
> > > 1, 2 and 4 are the ones defined in the paper. And, according to the reformulation of 2 (i.e., equivalence between Eq.9 and Eq.10), we can have 3 ($\Leftrightarrow$ 1) which is clearly equivalent to 4, resulting in 4$\Leftrightarrow$3$\Leftrightarrow$1; it shows the equivalence between 1 Pow-EMLR and 4 Pow-TMLR.
> > > It seems to be more straightforward than the authors' claim that considers the connection to SPD-MLR;
> > >
> > > 5. SPD-MLR': $\langle A, S^\theta-P\rangle$
> > >
> > > This paper discusses 5$\Leftrightarrow$2$\Leftrightarrow$1 in a bit complicated manner.
> > >
> > > ### 2.
> > > From Thm 2, it is understandable that SPD-MLR in Eq.16 is reduced to the above-mentioned 5 SPD-MLR' with Euclidean bias parameter P. It, however, is hard to find *theoretical* equivalence between the bias $\langle A,P\rangle$ and $b'$; it is easy to *re-parameterize* them, but are they intrinsically equivalent from the viewpoint of training (dynamics)?
> > > By seeing the performance gap between 4($\Leftrightarrow$3) and 1 (Table 3), I'm afraid that there might also be a gap between 2($\Leftrightarrow$5) and 1.
> > > This is the reason why I said "*it is necessary to compare the SPD-MLR with Tangent-MLR and practical EMLR*" in the review comment; namely, the performance comparison between 1 Pow-EMLR and 2 Pow-EMLR'.

---

> > > > ### Author Response · Authors · 2024-11-28
> > > > **Thanks for the further clarification and insightful comments! 😄**
> > > >
> > > > **Table A** Comparison of Pow-TMLR, Pow-EMLR, and Pow-EMLR' on all three  fine-grained datasets.
> > > > | Classifier |      Air      |               |     Birds     |               |      Car      |               |
> > > > |:----------:|:-------------:|:-------------:|:-------------:|:-------------:|:-------------:|:-------------:|
> > > > |            | Top-1 Acc (%) | Top-5 Acc (%) | Top-1 Acc (%) | Top-5 Acc (%) | Top-1 Acc (%) | Top-5 Acc (%) |
> > > > |  Pow-TMLR  |     69.58     |     88.68     |     52.97     |      77.8     |     51.14     |     74.29     |
> > > > |  (2) Pow-EMLR' |     73.03     |      90.4     |     63.96     |     85.02     |     80.06     |     94.02     |
> > > > |  (1) Pow-EMLR  |     72.07     |     89.83     |     63.29     |     84.66     |     80.43     |     94.15     |
> > > >
> > > > Thank you for the further clarification and insightful comments! We now fully understand your concern. To address this, we conducted additional experiments on Pow-EMLR (1) and Pow-EMLR' (2) across all three fine-grained datasets.
> > > >
> > > > The results are presented in Tab. A, where we also recap the performance of Pow-TMLR. We can observe that Pow-EMLR and Pow-EMLR' exhibit similar performance, indicating that $\langle A, P \rangle$ and $b^{\prime}$ are equivalent with respect to training dynamics. The marginal performance gap might be due to the increased number of parameters of $P$ compared to $b^{\prime}$.
> > > >
> > > > As discussed in the main paper, although Pow-TMLR differs from Pow-EMLR only in a constant transformation, the underlying mechanism is much different. As also mentioned in our response to the Reviewer $\textcolor{red}{2skh}$, we were indeed confused to see the difference between Pow-EMLR and Pow-TMLR at the very beginning. We had expected that the tangent classifier could explain the matrix power well, as they only differ in a constant transformation. However, the experimental results do not support our hypothesis.
> > > >
> > > > Delving deeper into this myth, we finally discovered that Pow-EMLR is implicitly a Riemannian MLR, which is a fundamentally different mechanism from the tangent perspective. This finding explains not only the mechanism underlying matrix functions in GCP but also the unexpected performance gap observed. We hope our findings will inspire further research of GCP from a geometric perspective.
> > > >
> > > > If you have any further comments, we would be delighted to discuss them.

---

> > > > > ### Comment · Reviewer_JvSJ · 2024-12-02
> > > > >
> > > > > Thanks a lot for the additional results!
> > > > > That's convincing evidence to support your claim.
> > > > > I hope you incorporate those results and discussions regarding linear classifiers into the manuscript and carefully revise the presentation to further clarify the claims, since it is a bit hard to grasp the contribution in the current manuscript.
> > > > >
> > > > > So, I happily increase my score.

---

> ### Author Response · Authors · 2024-12-02
> **Thanks! 😄**
>
> Thank you very much for the positive feedback and suggestive comments!
>
> These discussions will be shown in our final version, as we cannot upload pdf now. We will further revise our presentation to clarify our key contributions! 😄

---

### Official Review · Reviewer_SGk2 · 2024-11-03

**Soundness:** 3
**Presentation:** 3
**Contribution:** 3
**Rating:** 6
**Confidence:** 2

**Summary:**

This paper provides a theoretical understanding of matrix function normalization in Global Covariance Pooling (GCP) from the perspective of Riemannian geometry, specifically addressing the use of matrix logarithm and power for classifying covariance matrices. By interpreting these matrix functions through Riemannian and tangent classifiers, the authors reveal that the efficacy of GCP is rooted in its respect for the Riemannian structure of Symmetric Positive Definite (SPD) matrices. Extensive experiments demonstrate that matrix power, often more effective in practical applications, implicitly respects a Riemannian classifier, justifying its empirical performance in GCP.

**Strengths:**

- The paper offers a unified theoretical explanation of the working mechnism of matrix functions which demonstrates that matrix functions in GCP are implicitly Riemannian classifiers. Based on the unified theoretical analysis, it also shows that the empirical advantages of matrix power over matrix logarithm in the GCP could be attributed to the characteristics of the underlying Riemannian metrics.
- The paper experimentally validates that the mechanism of matrix normalization should be attributed to Riemannian classifiers instead of tangent classifiers across different datasets and networks.

**Weaknesses:**

- In sec. 5.3, the analysis shows that the matrix power function inherently offers better performance over the matrix logarithm based on its associated Riemannian metric ((θ, 1, 0)-EM). The description of (θ, 1, 0)-EM as a “balanced” alternative to LEM might be overly simplified. The analysis could benefit from a more nuanced discussion of the limitations of each metric, such as specific conditions where (θ, 1, 0)-EM may not perform as well as LEM.
- In sec. 6.1, although ScalePow-EMLR is claimed to be equivalent to Pow-EMLR under scaled settings, the analysis lacks a comprehensive justification for this equivalence. Exploring how different scaling factors impact performance would help.
- The paper primarily evaluates ResNet-based architectures. How well would the findings generalize to other architectures, such as ViT.

**Questions:**

See weakness

---

> ### Author Response · Authors · 2024-11-23
> **Response to Reviewer SGk2 (Part 1)**
>
> We thank Reviewer $\textcolor{purple}{SGk2}$ for the encouraging feedback and the constructive comments! In the following, we respond to the concerns point by point. 😄
>
> ***
> ## **1. Nuanced Discussion of LEM and PEM**
>
> **Notations.** $(\theta, 1, 0)$-EM refers to the Power-Euclidean Metric (PEM), while $(1, 0)$-LEM corresponds to the standard Log-Euclidean Metric (LEM). For simplicity, we use PEM and LEM in the following discussion.
>
> ### **Comparison of LEM and PEM**
>
> Below is a refined discussion of Sec. 5.3, as summarized in Tab. A.
>
> **Table A**: A summary of key differences between LEM and PEM.
> |                   |                   LEM                   |                   PEM                   |
> |:-----------------:|:---------------------------------------:|:---------------------------------------:|
> |   Limitation      | Overly Stretched Eigenvalues           | (Alleviated) Swelling Effect           |
> |     Geodesic completeness      | ✅                                    | ❌ |
> |     Relation      | N/A                                    | Approaching LEM as $\theta \rightarrow 0$ |
>
> - **Overly Stretched Eigenvalues in LEM.** The matrix logarithm in LEM can excessively stretch eigenvalues [a], which may lead to suboptimal results, especially in high-dimensional settings like those encountered in GCP. In contrast, PEM exhibits more moderate eigenvalue scaling. Tab. B provides a numerical illustration.
>
> **Table B**: A numerical comparison of log and square root functions. Given eigenvalues $a=10^3$ and $b=10^{-3}$, the log excessively compresses the magnitude.
> | Expression              | $\frac{a}{b}$  | $\frac{\log(a)}{\log(b)}$ | $\frac{\sqrt{a}}{\sqrt{b}}$ |
> |-----------------------------|----------------|---------------------------|-----------------------------|
> | Value                   | $1,000,000$    | $-1$                      | $1,000$                    |
>
> - **Swelling Effect in PEM.** PEM suffers from the swelling effect, which may affect certain applications, such as diffusion tensor imaging. While LEM does not exhibit swelling [b], PEM partially alleviates this issue but cannot eliminate it. Following [b], we analyzed the swelling effect on geodesic interpolation under different metrics. Tab. C shows that LEM demonstrates no swelling, while the swelling in PEM reduces as $\theta$ decreases.
>
> **Table C**: Determinants of geodesic interpolations on the $3 \times 3$ SPD manifold under PEM and LEM. The swelling effect is indicated by increasing determinant values. Here, $\theta$-PEM denotes PEM with power equal to $\theta$.
> | Interpolation Step  | 0    | 1       | 2       | 3       | 4       | 5       | 6       | 7       | 8       | 9       |
> |---------------------|------|---------|---------|---------|---------|---------|---------|---------|---------|---------|
> | 1.0-PEM        | 3.07 | 104.86  | 182.09  | 234.38  | 261.35  | 262.64  | 237.86  | 186.64  | 108.61  | 3.38    |
> | 0.5-PEM        | 3.07 | 18.67   | 39.93   | 59.53   | 71.96   | 73.79   | 64.14   | 45.01   | 21.73   | 3.38    |
> | 0.1-PEM       | 3.07 | 4.25    | 5.42    | 6.38    | 6.96    | 7.05    | 6.62    | 5.75    | 4.6     | 3.38    |
> | LEM            | 3.07 | 3.10    | 3.14    | 3.17    | 3.20    | 3.24    | 3.27    | 3.31    | 3.34    | 3.38    |
>
> - **Completeness.** As shown in [c, Tab. 2], LEM is geodesically complete, while PEM is incomplete. This limitation may affect applications requiring the Riemannian exponential map to be globally defined. However, this limitation does not impact GCP, as GCP does not require completeness.
>
> - **Relation Between PEM and LEM.** As shown in [d, Sec. 3.1], PEM converges to LEM as power $\theta \to 0$. Tab. C also demonstrates this, where smaller values of $\theta$ reduce swelling, making PEM resemble LEM. This suggests that PEM can be viewed as a balanced metric between LEM and the Euclidean Metric (EM).
>
> ### **Empirical Applications**
>
> The limitations of LEM and PEM depend on specific applications:
> - **High-dimensional SPD Matrices:** The eigenvalue stretching in LEM can make it less suitable for high-dimensional settings, such as those in GCP.
> - **Very Small SPD Matrices:** Conversely, the swelling effect in PEM may become unsatisfactory for very small SPD matrices, such as the $3 \times 3$ matrices used in diffusion tensor imaging.
>
> Overall, the choice of metric should be treated as a hyperparameter. Due to the non-convex nature of deep networks, it is challenging to identify a universally optimal metric across all tasks. Our work highlights the tradeoffs between LEM and PEM, providing a theoretical framework for practitioners to explore metrics tailored to specific applications.
>
> ***

---

> ### Author Response · Authors · 2024-11-23
> **Response to Reviewer SGk2 (Part 2)**
>
> ## **2. Comparison of Pow-EMLR, Pow-TMLR, and SaclePow-EMLR under different power factors.**
>
> **Table. D**: Results of different powers under the ResNet-50 backbone on the Aircrafts and Cars dataset.
> |     Classifier     |   Aircrafts   |               |      Cars     |               |
> |:------------------:|:-------------:|:-------------:|:-------------:|:-------------:|
> |                    | Top-1 Acc (%) | Top-5 Acc (%) | Top-1 Acc (%) | Top-5 Acc (%) |
> |    Pow-TMLR-0.25   |     65.41     |     86.71     |     41.47     |     66.66     |
> | ScalePow-EMLR-0.25 |     **72.76**     |     **90.31**     |     61.78     |     84.04     |
> |    Pow-EMLR-0.25   |     71.47     |     90.04     |     **62.88**     |     84.14     |
> |    Pow-TMLR-0.5    |      67.9     |     88.75     |     55.01     |     77.95     |
> |  ScalePow-EMLR-0.5 |     74.29     |     91.12     |     62.42     |     84.82     |
> |    Pow-EMLR-0.5    |     74.17     |     91.21     |     62.83     |     84.85     |
> |    Pow-TMLR-0.7    |     65.92     |     87.49     |     50.68     |     74.12     |
> |  ScalePow-EMLR-0.7 |     **74.26**     |     **91.15**     |     **64.22**     |     **83.67**     |
> |    Pow-EMLR-0.7    |     74.17     |     90.49     |     61.41     |     82.39     |
>
> Following your suggestion, we conducted additional experiments with varying power factors on the Aircrafts and Cars datasets, i.e., $\theta = 0.25, 0.5, 0.7$. For these experiments, we employed accurate Singular Value Decomposition (SVD) for calculating the matrix power and used Padé approximants for backpropagation [e]. The results are summarized in Tab. D. Note that, since we used SVD to compute all the matrix power in these experiments, the results for $\theta = 0.5$ might differ from those reported in Tab. 5 of the main paper.
>
> From the results, it is clear that Pow-EMLR consistently achieves performance comparable to ScalePow-EMLR while outperforming Pow-TMLR across all power factors. This provides further evidence supporting our claim that the mechanism underlying matrix functions in GCP should be attributed to the Riemannian MLRs they implicitly respect.
>
> ***Remark:** The above has been added to Sec. 6.2 in our revised paper.*
>
> ***
> ## **3. Experiments on the Second-order Transformer (SoT)**
>
> **Table E**: Comparison of Pow-EMLR against Pow-TMLR under the SoT-7 backbone on the ImageNet-1k dataset.
> | Classifier | Top-1 Acc (%) | Top-5 Acc (%) |
> |:----------:|:-------------:|:-------------:|
> |  Pow-TMLR  |     75.79     |     92.91     |
> |  Pow-EMLR  |     **76.11** |     **93.05** |
>
> We appreciate the reviewer’s suggestion to evaluate our findings on ViT. Following [f], we conduct experiments using the Second-order Transformer (SoT) [g] on the ImageNet-1k dataset. Specifically, we use the 7-layer SoT (SoT-7) architecture and train the model up to 250 epochs with a batch size of 512, keeping other settings consistent with [f]. The results are presented in Tab. E.
>
> Pow-EMLR still outperforms Pow-TMLR under the SoT-7 backbone. These results further support our claim that tangent classifiers cannot adequately explain the matrix functions used in GCP, while the underlying mechanism can be better explained by our Riemannian perspective.
>
> ***Remark:** The above has been added into Sec. H.3 in our revised paper.*
>
> ***
> ## References
> > [a] Why approximate matrix square root outperforms accurate SVD in global covariance pooling
> >
> > [b] Log‐Euclidean metrics for fast and simple calculus on diffusion tensors
> >
> > [c] RMLR: Extending multinomial logistic regression into general geometries
> >
> > [d] The geometry of mixed-Euclidean metrics on symmetric positive definite matrices
> >
> > [e] Why approximate matrix square root outperforms accurate SVD in global covariance pooling?
> >
> > [f] Fast differentiable matrix square root
> >
> > [g] SoT: Delving Deeper into Classification Head for Transformer

---

> > ### Comment · Reviewer_SGk2 · 2024-11-26
> > **Reply by  Reviewer SGk2**
> >
> > Thank you for your detailed reply.
> >
> > My concerns have been largely addressed.

---

> > > ### Author Response · Authors · 2024-11-26
> > > **Thanks**
> > >
> > > Thanks for the reply! We appreciate your time and efforts during the review and discussion. 😄

---

> ### Author Response · Authors · 2024-11-27
> **Additional Experiments on Second order Transformer (SoT)**
>
> **Additional Experiments on Second-order Transformer (SoT)**
>
> Dear reviewer, we have trained SoT-7 with ScalePow-EMLR to further validate our claim. The following table renews the previous Tab. E. Pow-EMLR still achieves similar performance to ScalePow-EMLR, but outperforms Pow-TMLR. These results further support our claim that tangent classifiers cannot adequately explain the matrix functions used in GCP, while the underlying mechanism can be better explained by our Riemannian perspective.
>
> We have revised our manuscript accordingly.
>
> **Table F**: Comparison of Pow-EMLR, ScalePow-EMLR and Pow-TMLR under the SoT-7 backbone on the ImageNet-1k dataset.
> |   Classifier  | Top-1 Acc (%) | Top-5 Acc (%) |
> |:-------------:|:-------------:|:-------------:|
> |    Pow-TMLR   |     75.79     |     92.91     |
> | ScalePow-EMLR |     **76.14**    |     **93.18**     |
> |    Pow-EMLR   |     76.11     |     93.05     |

---

### Meta-Review · Area_Chair_o9Mb · 2024-12-20

**Metareview:**

This work aims to provide rigorous understanding on the normalisation of SPD matrix, a feature representation resulted by applying global covariance pooling in deep neural networks. In particular, it focuses on understanding the association between matrix normalisation and the classifiers applied to the resulted feature representation. Both theoretical analysis and experimental investigations are conducted to serve the purpose. Reviewers comment that the theoretical stuffs are insightful and the empirical study is interesting and that the study is technically sound and the paper is well written. At the same time, the reviewers request more discussions, the clarification on claims, results, and the practical significance of this study, and so on. The authors provide the rebuttal and have good discussions with some reviewers. The rebuttal is informative and effectively addresses most of the concerns. All the final ratings are on the positive side. By checking the submission, the reviews, and the rebuttals, AC agrees with the reviewers on their observations and recommends this work to be accepted. In addition, the authors shall properly incorporate the answers in the rebuttal into the manuscript and further improve the presentation.

**Additional Comments On Reviewer Discussion:**

Reviewers comment that the theoretical stuffs are insightful and the empirical study is interesting and that the study is technically sound and the paper is well written. At the same time, the reviewers request more discussions, the clarification on claims, results, and the practical significance of this study, and so on. The authors provide the rebuttal and have good discussions with some reviewers. The rebuttal is informative and effectively addresses most of the concerns. By checking the submission, the reviews, and the rebuttals, AC agrees with the reviewers on their observations and recommends acceptance. Also, AC suggests that the authors properly incorporate the answers in the rebuttal into the manuscript and further improve the presentation.

---

### Decision · Program_Chairs · 2025-01-22

Accept (Poster)